# Adult-neurogenesis allows for representational stability and flexibility in early olfactory system

Zhen Chen[1], Krishnan Padmanabhan[2]*

[1]Department of Brain and Cognitive Sciences, University of Rochester, Rochester, United States; [2]Department of Neuroscience, University of Rochester School of Medicine and Dentistry, Rochester, United States

## eLife Assessment

This paper presents an **important** theory and analysis of the role of neurogenesis and inhibitory plasticity in the drift of neural representations in the olfactory system. For one of the findings, regarding the impact of neurogenesis on the drift, the evidence remains **incomplete**. The reason lies in the differences in variability/drift of the mitral/tufted cell responses observed in the model compared to experimental observations, where these responses remain stable over extended time scales.

**\*For correspondence:**
Krishnan_Padmanabhan@urmc.rochester.edu

**Competing interest:** The authors declare that no competing interests exist.

**Abstract** In the olfactory system, adult-neurogenesis results in the continuous reorganization of synaptic connections and network architecture throughout the animal's life. This poses a critical challenge: How does the olfactory system maintain stable representations of odors amidst this ongoing circuit instability? Utilizing a detailed spiking network model of early olfactory circuits, we uncovered dual roles for adult-neurogenesis: one that both supports representational stability to faithfully encode odor information, and also one that facilitates plasticity to allow for learning and adaptation. In the main olfactory bulb, adult-neurogenesis affects neural codes in individual mitral and tufted cells but preserves odor representations at the neuronal population level. By contrast, in the olfactory piriform cortex (PCx), both individual cell responses and overall population dynamics undergo progressive changes due to adult-neurogenesis. This leads to representational drift, a gradual alteration in stimulus-evoked activity patterns. Both processes are dynamic and depend on experience such that repeated exposure to specific odors reduces the drift due to adult-neurogenesis; thus, when the odor environment is stable over the course of adult-neurogenesis, it is spike-timing-dependent plasticity that leads representations to remain stable in the PCx; when those olfactory environments change, adult-neurogenesis allows cortical representations to track environmental change. Whereas perceptual stability and plasticity due to learning are often thought of as two distinct, often contradictory processes in neuronal coding, we find that adult-neurogenesis serves as a shared mechanism for both. In this regard, the quixotic presence of adult-neurogenesis in the mammalian olfactory bulb that has been the focus of considerable investigation in chemosensory neuroscience may be the mechanistic underpinning behind an array of complex computations.

## Introduction

In the rodent brain, the main olfactory bulb (MOB) is one of two regions where neurogenesis persists throughout the animal's lifetime (*Lledo et al., 2006*). Adult-born cells in the subventricular zone migrate to the MOB, where nearly 95% differentiate into inhibitory adult-born granule cells (abGCs) forming reciprocal dendro-dendritic connections (*Urban and Sakmann, 2002*; *Schoppa and Urban,*

*2003*) with the principal excitatory cells of the MOB, the mitral and tufted (M/T) cells. M/T cells in turn relay the chemosensory information to the downstream targets including the olfactory or piriform cortex (PCx) (*Sosulski et al., 2011*; *Ghosh et al., 2011*; *Miyamichi et al., 2011*). Although both the cellular composition and synaptic organization of this circuit undergo constant changes due to adult-neurogenesis (*Lepousez et al., 2013*), animals are nonetheless able to perform incredibly complex behavioral tasks related to the determination of odor identity and concentration, requiring that they maintain stable representations of odors. This raises two critical questions: Does the encoding of odors via patterns of neural activity throughout the early olfactory system undergo changes as a result of this ongoing plasticity, and if so, how? If changes in neural activity due to adult-neurogenesis percolate throughout early olfactory circuits, what computational principles allow the system to represent the same odor stably over time?

Previous studies have explored the plasticity at synapses between M/T cells and abGCs, particularly in relation to olfactory learning (*Lepousez et al., 2014*; *Livneh and Mizrahi, 2012*; *Nissant et al., 2009*). It is known that sensory experience influences the survival and synaptic turnover of abGCs with M/T cells (*Lepousez et al., 2013*; *Lledo and Valley, 2016*), impacting both the normal activity of these cells (*Breton-Provencher et al., 2009*) and animals' performance in odor discrimination tasks (*Kouremenou et al., 2020*). However, the implications of these local synaptic changes in the bulb on the downstream areas, including in the PCx, which is critical for olfactory perception and learning (*Bekkers and Suzuki, 2013*; *Choi et al., 2011*), remain largely unexplored.

The three-layer PCx is thought to be a region where the components of an odor are assembled into an olfactory percept or representation. Subsets of piriform cells (PCs) respond when multiple M/T cells are activated in complex temporal sequences (*Smear et al., 2013*; *Chong et al., 2020*; *Gill et al., 2020*). A given population of M/T cells from a single glomerulus, a neuropil-like structure where inputs from individual olfactory receptor neuron types converge (*Mombaerts et al., 1996*; *Buck and Axel, 1991*), project randomly to a distributed array of PCs (*Sosulski et al., 2011*; *Ghosh et al., 2011*; *Miyamichi et al., 2011*). As odor information in PCx is coded for through the combinatorial activity of PCs (*Davison and Ehlers, 2011*; *Stern et al., 2018*), it is hypothesized that PCx is where stable representations of odors are maintained (*Pashkovski et al., 2020*). However, recent evidence is beginning to challenge this idea (*Schoonover et al., 2021*). First, NMDA-mediated long-term potentiation (LTP) in both afferent and associative fibers to PCx (*Kanter and Haberly, 1990*; *Jung and Larson, 1994*) is similar to that observed in the CA1 region of the hippocampus. PCs thus undergo synaptic plasticity on scales observed in other regions where representations are not stable (*Ziv et al., 2013*). Second, recent studies examining the responses of PCs over months demonstrate that patterns of activity in both individual cells and populations of cells can drift over time (*Schoonover et al., 2021*). This representational drift has led to the hypothesis that PCx, rather than being a primary sensory area, may be more like an associative cortical region (*Haberly and Bower, 1984*; *Haberly, 2001*). However, the mechanisms by which this drift in PCx occurs remain unknown.

One clue that adult-neurogenesis in MOB may be the mechanism underlying these diverse processes is that neurogenesis occurs on a similar time scale to the representational drift in PCx, and the turnovers of abGCs are highly experience dependent (*Livneh and Mizrahi, 2012*), which is also reflected in the PCx drift (*Schoonover et al., 2021*). More frequent exposure to the same odor slows down the representational drift in PCx, while the drift rate increases again once this frequent exposure is halted (*Schoonover et al., 2021*).

We therefore hypothesized that the long-term plasticity conferred by adult-neurogenesis in MOB percolates through the olfactory system, serving as the mechanism of representational drift in PCx. To test this hypothesis, we used a spiking neuronal network model that replicated the circuit architecture within and between the MOB and PCx, integrating both the ongoing network restructuring due to adult-neurogenesis and the short-term plasticity of abGCs. We found that adult-neurogenesis differentially modulated odor responses and representations in M/T cells in the MOB and PCs in PCx. While the MOB maintained stable population representations despite changes in individual M/T cell responses due to neurogenesis, the PCx shows variation at both individual and population levels, leading to representational drift. Furthermore, when we incorporated spike-timing-dependent plasticity (STDP) at the synapses on abGCs, repeated odor exposure stabilized the PCx representations, resulting in a reduction in drift rate. Taken together, we identified how the rules of plasticity on short-term time scales such as STDP and long-term time scales like adult-neurogenesis allow networks to

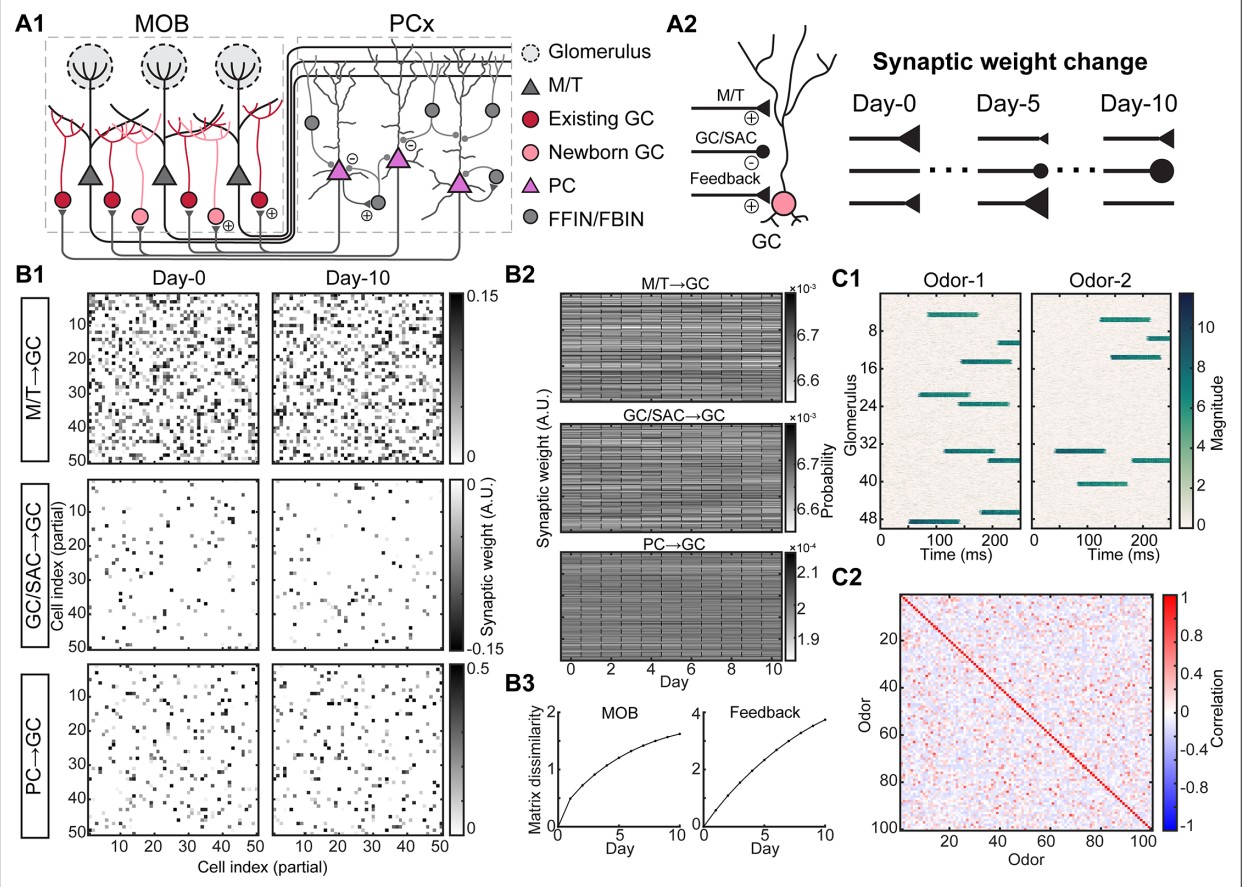

**Figure 1.** Spiking network model of adult-neurogenesis. (**A1**). Schematic of early olfactory circuit. M/T: mitral/tufted cells, GC: granule cells, abGCs: adult-born GCs, PCs: piriform cortical cell, FFIN: feedforward inhibitory neuron, FBIN: feedback inhibitory neuron. Example plus signs indicate excitatory synapses and minus signs indicate inhibitory synapses. (**A2**). Schematic of synaptic reshuffling of GCs due to adult-neurogenesis. On each day, 10% of GCs have their weights reshuffled, including the synaptic weights from M/T cells, other GCs or short-axon cells (SACs), and PCs to GC (feedback). (**B1**). Partial weight matrix on two example days between GCs and M/T cells (top), other GCs or SACs (middle), and feedback from PCs (bottom). (**B2**). Histograms of all synaptic weights across days. (**B3**). Weight matrix dissimilarity between Day-0 and each other day. (**C1**). An example model odor generated by stimulating different combinations of glomerular identity and timing of activation. (**C2**). Pair-wise correlations between model odors show the relative similarities and differences across all model odors.

The online version of this article includes the following figure supplement(s) for figure 1:

**Figure supplement 1.** Odor definition and synaptic weight changes.

both preserve sensory representations in some circuits like the MOB, while allowing other representations to change like those in PCx. Our work reveals the nuanced role played by adult-neurogenesis in balancing stability and adaptability in the chemosensory system.

## Results
### Spiking network model of adult-neurogenesis in early olfactory system

To understand the functional role of adult-neurogenesis of granule cells (GCs) on the odor representations in both MOB and PCx, we used a detailed spiking neural model that recapitulates the circuit architecture and neural dynamics of the early olfactory system (*Figure 1A1*, Methods). To this, we added adult-neurogenesis, which we modeled as a process where a subset of GCs are killed off and new GCs are integrated into the network as abGCs (*Figure 1A1*; *Lepousez et al., 2013*; *Aimone and Gage, 2011*). We modeled this process by reshuffling the synaptic weights of 10% of GCs in the network with the total number of GCs fixed. The reshuffled synaptic weights include the weights between these abGCs and M/Ts, abGCs and other GCs or short axon cells (SACs), and the

top-down feedback projections from PCs to abGCs (*Figure 1A2*). These new synaptic weights for abGCs were sampled from the same distributions that generated the initial network model, such that although individual weight values and connectivity were being modulated, the total distribution of synaptic weights remain stable (*Figure 1B2*). We simulated a total of 11 days of neurogenesis from Day-0 and Day-10 by which point almost all GCs in the model have gone through adult-neurogenesis. Over the course, the synaptic weights between M/Ts and GCs, GCs/SACs, and GCs, PCs and GCs were changed to match the observed changes in experiments (*Figure 1B1–B3*, *Figure 1—figure supplement 1*). Although the empirical rate of adult-neurogenesis has been found to vary in rodents depending on the measurement approach and the behavioral paradigms being used (*Pignatelli and Belluzzi, 2010*), we selected a rate of 10% for a reasonable computational time. Notably, changing this rate in our model only affected the scale of the observations without qualitatively changing the core results. These gradual changes in the network connectivity allowed us to probe the long-term effect of adult-neurogenesis on odor processing and the effect it had on modulating odor representations in MOB and PCx over time.

Odors drive coordinated activity in structures called glomeruli from which M/T cells receive their information about the odors. To match the glomerular activation patterns both in terms of the identity and onset timing in response to real odors (*Fantana et al., 2008*; *Gschwend et al., 2016*; *Vincis et al., 2012*), we generated a panel of 100 model odors where each one activated 6–20% of the total 50 glomeruli in the model with different onset latencies (*Figure 1C1*, *Figure 1—figure supplement 1*). Each model odor was presented during a 250-ms time window (4 Hz sniff), corresponding to a single sniff that is ethologically and behaviorally relevant time-scale for rodents (*Rinberg et al., 2006*; *Uchida and Mainen, 2003*; *Wesson et al., 2008*). Our generated panel of model odors spanned a wide range of similarities, ranging from distinct odor pairs with low pairwise correlations (see Methods) to highly similar odor pairs with high pairwise correlations (*Figure 1C2*). The input to our network, the model odors at the glomerular level, remained unchanged across simulated days consistent with previous observations of stable odor responses of glomeruli (*Kato et al., 2012*). Using this experimental paradigm, we could study how the same odor inputs were differently encoded over the course of adult-neurogenesis by both the principal cells in the MOB and the PCx (*Bekkers and Suzuki, 2013*; *Uchida et al., 2014*).

## Modulation of individual cell responses and preservation of population representations in MOB

First, we examined how adult-neurogenesis affected the responses of both individual M/T cells in the bulb and the population. We found that individual M/T cells changed their responses to the same odor across days due to adult-neurogenesis, with some cells decreasing the firing rate responses (*Figure 2A1*, top) while other cells increased the magnitude of their responses (*Figure 2A2*, bottom, *Figure 2—figure supplement 1*), similar to what has been observed by others using calcium imaging of M/T cell activity (*Kato et al., 2012*; *Yamada et al., 2017*). These changes in gain are consistent with a model of normalization, wherein the firing of an individual neuron is modulated within a regime. Notably, individual M/T cell responses also exhibited some extent of variability across repeated presentations of the same odor within a single session (*Figure 2—figure supplement 1*, error bars). This within-session variability occurs on a much shorter time scale than adult-neurogenesis and therefore cannot be attributed to it. We return to this point and its implications in the Discussion. However, when we examined the M/T responses to the same odor across the population (*Figure 2B2*), we found that the overall pattern of population responses was preserved across days despite the ongoing neurogenesis. In this example, the odor activated 6 of the 50 glomeruli. Although individual M/T responses changed (*Figure 2A1, A2* arrows), the groups of M/T cells driven by the same glomerulus preserved the temporal structure of their firing as a whole. Each individual M/T cell received a certain amount of inhibition from a subset of GCs. With the incorporation of abGCs that eliminated some connections and randomly established new ones, the synaptic weights of abGCs changed the inhibition at the individual M/T cell level but preserved the distribution of inhibition across the population of M/T cells associated with each glomerulus. As a result, adult-neurogenesis effectively preserved the ensemble response to each odor for M/T cells associated with a single glomerulus but led to a reshuffling of the identity of which M/T cells responded at what time epoch and with what magnitude.

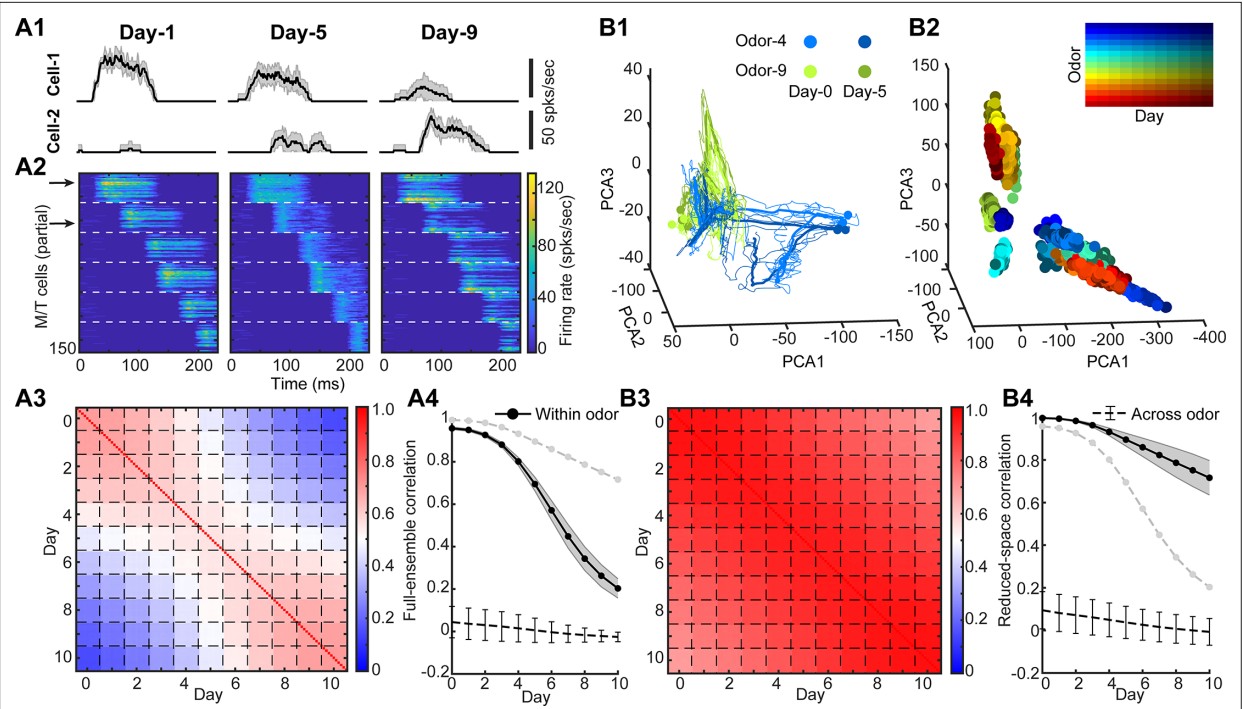

**Figure 2.** Adult-neurogenesis modulates individual M/T cell responses but preserves population representation. (**A1**) Trail-averaged firing rate of two example M/T cells responding to the same odor (within-odor) on three different days (mean ± SD, $n = 10$ trials). (**A2**) Firing rate patterns of odor-activated M/T cells to the same odor on 3 days. M/T cells driven by different glomeruli are separated by a white dashed line. The two arrows correspond to the two example cells in (**A1**). (**A3**) Pairwise within-odor full-ensemble correlation of single-trial M/T cell responses between each day averaged across all odors ($n = 100$). Each small box separated by the dashed lines is a $10 \times 10$ matrix ($n = 10$ trials) corresponding to autocorrelation (same day on diagonal) and cross-correlation (different days off diagonal). (**A4**) Full-ensemble correlation of trial-averaged M/T cell responses for within-odor (black solid line with error bar: mean ± SD, $n = 100$ odors) and across-odor (black dashed line with error bar: mean ± SD, $n = 10$ pairs) between Day-0 and each other day. Gray dashed line with no error bar: same as the black solid line in (**B4**). (**B1**) Low-dimensional trajectories of M/T responses to two example odors (color) on two different days (darkness of color). Thin curves: single-trial trajectories, thick curves: trial-averaged trajectories, points: maximal-distance points on trajectory to the origin. (**B2**) Only maximal-distance points are shown for different odors on different days. (**B3**) Similar to (**A3**) but for reduced-space correlation computed using the low-dimensional M/T trajectories (single-trial). (**B4**) Similar to (**A4**) but for reduced-space correlation computed using the low-dimensional M/T trajectories (trial-averaged). Gray dashed line with no error bar: same as the black solid line in (**A4**).

The online version of this article includes the following figure supplement(s) for figure 2:

**Figure supplement 1.** Example M/T cell responses on Day-0 and Day-10.

Next, we wanted to quantify the changes in M/T odor response across days due to adult-neurogenesis. We took the odor-evoked firing rate over the 250-ms window across all M/T cells as a high-dimensional ensemble vector and computed their correlation between Day-0 and each other day. We named this correlation as '*full-ensemble correlation*'. Regardless of using either single-trial (*Figure 2A3*) or trial-averaged responses (*Figure 2A4*), the within-odor full-ensemble correlation decreased significantly over time. This decrease is consistent with experimental observations (*Yamada et al., 2017*). The across-odor correlations, computed from responses evoked by different odors, remained low across all days (*Figure 2A4*). Therefore, these results indicated that the gradual changes in the odor-evoked responses of individual M/T cells (*Figure 2A1*) accumulated over time.

However, as we observed above, the overall pattern of M/T population responses was preserved (*Figure 2A2*), suggesting that the covariance, a measure of the shared fluctuations across M/T population, might remain stable. To test this, we projected M/T population responses into a low-dimensional space (*Figure 2B1, B2*, see Methods), where the time-varying odor-evoked responses across the population constituted a trajectory in the odor representation space of M/T cell population activity. For the two example odors, the M/T trajectories of the same odor stayed quite close on the two example days (*Figure 2B1*). If we simplified the single-trial trajectories by only plotting the maximal-distance points to the origin (*Figure 2B2*), we found that the points of the same odor (color) stayed

clustered together in the space across all days (darkness of the color) despite the ongoing circuit restricting due to adult-neurogenesis. The correlation of the low-dimensional trajectories, which we named as '*reduced-space correlation*', remained high across all days (*Figure 2B3*), and only dropped to ~0.7–0.8 on Day-10 (*Figure 2B4*). This is consistent with a recent experimental study using calcium-imaging on M/T cells (*Shani-Narkiss et al., 2023*). Additionally, while the authors (*Shani-Narkiss et al., 2023*) observed relative stability of odor responses in some of the individual M/T cells, they also observed that the full-ensemble showed a clear monotonic increase in response differences over long time intervals of months. In this regard, our results showed that although adult-neurogenesis gradually varied the responses of individual M/T cells, the low-dimensional representations of M/T cells in the reduced space remained stable across days despite the changes in local synaptic organization due to adult-neurogenesis.

## Modulation of both individual cell responses and population representations in PCx

M/T cells in MOB send random projections to PCx, such that each individual PC receives input from a random subset of M/T cells associated with different glomeruli (*Sosulski et al., 2011*; *Miyamichi et al., 2011*). PCx cells incorporate this feedforward information about odor identity and concentration from the individual M/T cells in the MOB (*Bekkers and Suzuki, 2013*) and assembles those components into an olfactory percept (*Davison and Ehlers, 2011*; *Pashkovski et al., 2020*). Although perceptual stability is a hallmark of sensory processing and the PCx has historically been thought to be a place where perceptual stability is established in the population code, recent evidence has started to challenge this idea (*Schoonover et al., 2021*). We hypothesized that the gradual changes in the individual M/T responses, passed through the nonlinearities of synaptic integration, would affect the responses of PCs at both individual and population levels. We studied this by looking at the activity of individual PCs and the ensembles of PCs in PCx.

Like M/T cells, the responses of individual PCs to the same odor varied across days. Some PCs increased their responses while others decreased responses (*Figure 3A1*, *Figure 3—figure supplement 1*), consistent with recent chronic recordings in PCx (*Figure 3—figure supplement 2*; *Schoonover et al., 2021*). However, unlike M/T cells, the overall pattern of PCs' responses varied as a result of adult-neurogenesis (*Figure 3A2*). Some PCs responding strongly on Day-1 became inactive on Day-9, whereas some PCs that did not respond on Day-1 became strongly activated on Day-9. Consequently, the combinatorial patterns of activated PCs by the same odor were reorganized by adult-neurogenesis on different days. Additionally, we also found that the temporal structure of firing rate in individual PCs was highly variable. For example, some PCs changed the onset and duration of their activation over the course of neurogenesis (*Figure 3A2*). As a consequence, the within-odor full-ensemble correlation showed a significant drop for PCs (*Figure 3A3, A4*). This is consistent with recent studies examining the responses of PCs over time and demonstrating that patterns of PCs' responses in both individual cells and populations of cells can *drift* over time (*Schoonover et al., 2021*).

Changes in the firing rates reflect the changes in the variance of individual cells (*Figure 3—figure supplement 3*), but they do not say anything about the covariance, nor do they capture the nonlinear interactions that shape the representations in PC populations when synaptic information is integrated across M/T cell inputs and passes through the threshold nonlinearity in individual PCs reflective of this integration. To study this, we calculated the covariance across population of PCs was impacted by adult-neurogenesis. In contrast to the M/T population where the representational trajectories of the same odor followed one another closely in the space across different days; we found that the low-dimensional trajectories of PCs showed a large variability across different days (*Figure 3B1*). As a result, the maximal-distance points for the trajectories on each day were broadly spread in the space across progressive days of neurogenesis (*Figure 3B2*). These results are consistent with representational drift in the odor encoding space. Quantitatively, the within-odor reduced-space correlation decreased substantially across days (*Figure 3B3, B4*). Our data suggest that as a consequence of both the random projections of M/T cells to different PCs and the ways in which the cortical cells integrate those inputs, PCx representations drift with ongoing adult-neurogenesis in the bulb.

Next, we asked how adult-neurogenesis geometrically reshaped the odor manifold and representational trajectories in the high-dimensional space? Recent computational models that have been

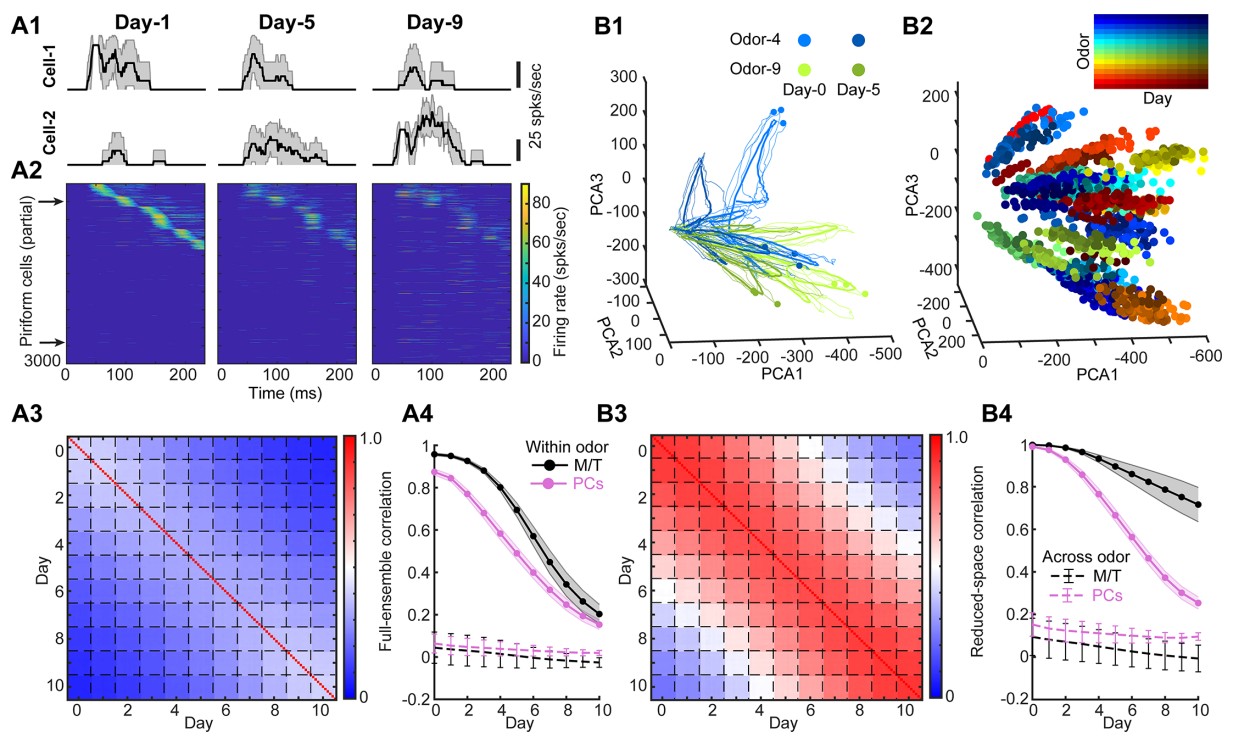

**Figure 3.** Adult-neurogenesis modulates both individual PCs' responses and population representations. (**A1**). Trail-averaged firing rate of two example PCs responding to the same odor on three different days (mean ± SD, *n* = 10 trials). (**A2**). Firing rate patterns of partial PCs responding to the same odor on 3 days. The two arrows correspond to the two example cells in (**A1**). (**A3**) Pairwise within-odor full-ensemble correlation of single-trial PCs' responses between each day averaged across all odors (*n* = 100). Each small box separated by the dashed lines is a 10 × 10 matrix (*n* = 10 trials) corresponding to autocorrelation (same day on diagonal) and cross-correlation (different days off diagonal). (**A4**). Full-ensemble correlation of trial-averaged cell (PCs: purple; M/T: black) responses for within-odor (solid line with error bar: mean ± SD, *n* = 100 odors) and across-odor (dashed line with error bar: mean ± SD, *n* = 10 pairs) between Day-0 and each other day. The curves for M/T cells are the same lines as in *Figure 2*. (**B1**) Low-dimensional trajectories of PCs' responses to two example odors (color) on two different days (darkness of color). Thin curves: single-trial trajectories, thick curves: trial-averaged trajectories, points: maximal-distance points on trajectory to the origin. (**B2**) Only maximal-distance points are shown for different odors on different days. (**B3**) Similar to (**A3**) but for reduced-space correlation computed using the single-trial principal component analysis (PCA) trajectories of PCs. (**B4**). Similar to (**A4**) but for reduced-space correlation computed using the trial-averaged PCA trajectories of PCs.

The online version of this article includes the following figure supplement(s) for figure 3:

**Figure supplement 1.** Example piriform cell responses on Day-0 and Day-10.

**Figure supplement 2.** Example piriform cell responses (trial-averaged) on all 10 days.

**Figure supplement 3.** Cell response changes between Day-0 and Day-10.

**Figure supplement 4.** Odor manifold and geometric reshaping of odor representations in PCx.

**Figure supplement 5.** Quantifying geometric reshaping and decoding analysis on M/T and PCs.

built on longstanding frameworks of population coding suggest that neural representations reside in a high-dimensional manifold (*Langdon et al., 2023*; *Kriegeskorte and Wei, 2021*). First, we reasoned that the structure of the odor manifold would be defined as the aggregation of the odor-evoked temporal responses within a sniff cycle to all the odors based on evidence showing that different odors evoke complex and distinct temporal activities in both M/T cells (*Baker et al., 2019*; *Gire et al., 2013*; *Schaefer and Margrie, 2007*) and PCs (*Rennaker et al., 2007*; *Chen and Padmanabhan, 2022*; *Haddad et al., 2013*). For visualization, we plotted the odor manifold in the three-dimensional principal component analysis (PCA) space (*Figure 3—figure supplement 4*). We found the odor manifolds of M/T cells were highly overlapping with each other, whereas for PCs they were much more separated. Across individual odors, we quantified the geometrical reshaping of representational trajectories by calculating the cosine similarity (see **Methods**), which measures the similarity between two vectors. It has a value of 1 if the two vectors have the same direction, and a value of 0 if they are orthogonal. We found that cosine similarity using the population firing rate of M/T cells and PCs had

a similar degree of decrease as a function of intervals (i.e., number of days between two representations) (*Figure 3—figure supplement 5A1*). However, the cosine similarity using representational trajectories remained stable for M/T cells but reduced for PCs (*Figure 3—figure supplement 5B1*).

If representations drift as a result of adult-neurogenesis, then behaviors too should be impacted, drifting in a similar timescale. To unpack this connection, we used the K-nearest neighbor algorithm as the decoder (see **Methods**). We found that the decoding accuracy for discriminating against two different odors using M/T cell population responses was high, but accuracy in PCs dropped substantially, capturing the representational flexibility in PCx (*Figure 3—figure supplement 5*).

## Experience-dependent plasticity enhances representational stability in PCx

So far, we have shown that adult-neurogenesis, a process that changes the circuit structure of the early olfactory system, is one mechanism by which representational drift in PCx happens. Interestingly, it has been reported that repeated experience of an odorant stabilizes odor representations and thus reduces the drift in PCx (*Schoonover et al., 2021*). The mechanism that drives drift also appears to stabilize that drift when the olfactory environment is stable, suggesting that a core feature of the mechanism should be plasticity. Since the synapses of abGCs have been shown to be highly plastic and experience-dependent (*Lepousez et al., 2014*; *Livneh and Mizrahi, 2012*; *Wu et al., 2020*),

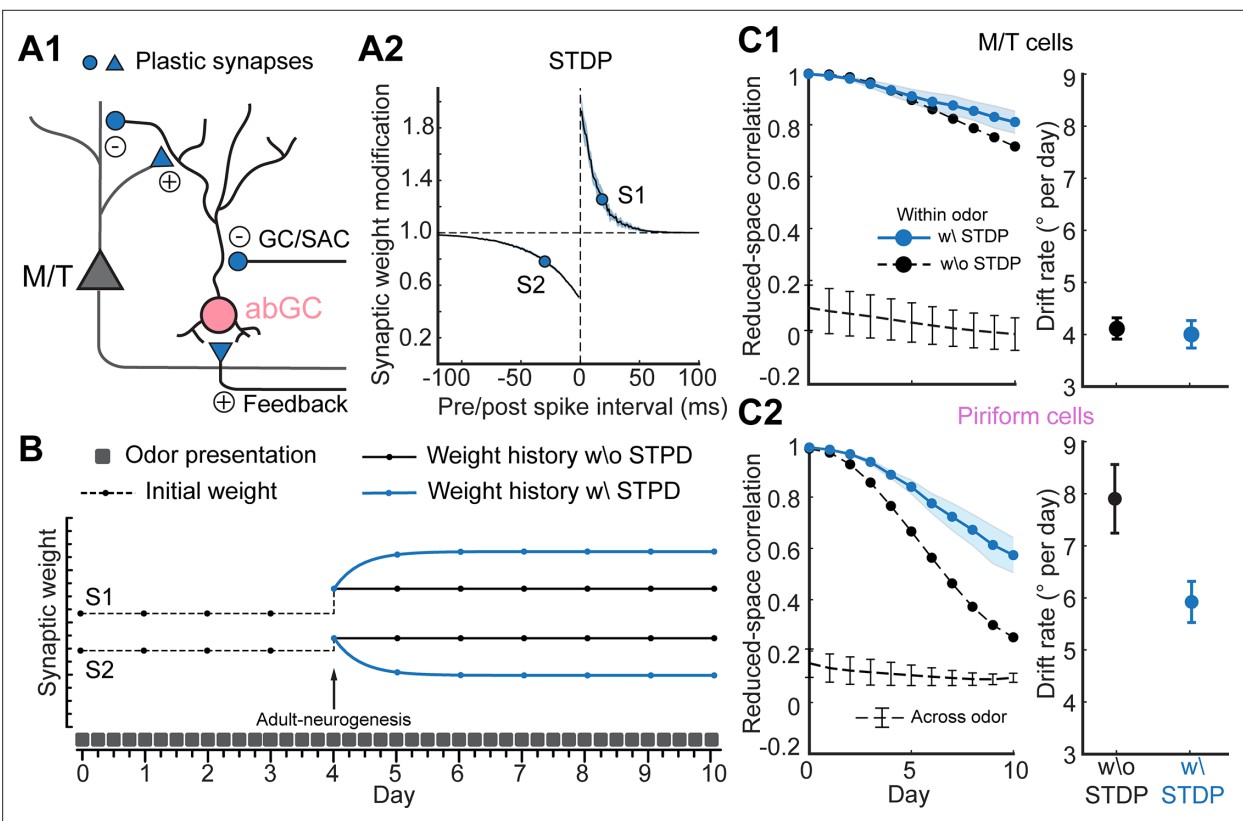

**Figure 4.** Experience-dependent plasticity enhances representational stability in PCx. (**A1**) Different types of plastic synapses (blue) related to an abGC. Circle: inhibitory; triangle: excitatory. (**A2**) Dependence of synaptic modification on pre/post inter-spike interval used for the spike-timing-dependent plasticity (STDP). S1 and S2 are two example synapses with different modification – S1: facilitated; S2: suppressed. (**B**) Weight history of the two example synapses in A2 when an odor is repeatedly presented. Adult-neurogenesis happens on Day-4 which randomly resets the weights. The synaptic weights stay constant without plasticity (black) while change trial by trial with plasticity (blue). (**C1**) Left: reduced-space correlation of trial-averaged M/T trajectories between Day-0 and each day. Blue solid line with error bar: within-odor and with STDP (mean ± SD, n = 10 odors); black dashed line without error bar: within-odor and without STDP. Black dashed line with error bar: across-odors, mean ± SD, n = 10 pairs. Right: drift rate for M/T trajectories with and without STDP. (**C2**) Same as C1 but for PCs trajectories.

The online version of this article includes the following figure supplement(s) for figure 4:

**Figure supplement 1.** Synaptic weight history of spike-timing-dependent plasticity (STDP).

we reasoned that experience-dependent plasticity of the same abGCs that drive representational in PCx may also stabilize that drift in response to the experience of encountering a stable olfactory environment.

To model experience-dependent changes at the level of synapses, we implemented an STDP rule (*Froemke and Dan, 2002*; *Bi and Poo, 2001*; *Markram et al., 2012*) across the diversity of the synapses related to an abGC (*Figure 4A1*). Depending on the spike timing interval of pre- and post-synaptic neurons, the synaptic weight could be either facilitated or suppressed (*Figure 4A2*, see Methods). Experience then was modeled as repeated presentations of the same set of odors every day (50 trials for each odor per day) to our network model, while adult-neurogenesis was occurring as previously described, with the network presumably learning these stable representations in the pattern of synaptic connectivity. In this example GC, the adult-neurogenesis that took place on Day-4 resulted in a new cell being added to the circuit, while the existing cell was removed, a process that randomly reset a group of synaptic weights (say, S1 and S2, *Figure 4B*) across two days of neurogenesis. Without STDP, the synaptic weights S1 and S2 would keep constant or fluctuate by some stochastic process, regardless of what was happening in the environment, including the effects of the repeated experience of an odor (black line, *Figure 4B*). The implementation of STDP meant that the synaptic weight may change on a trial-by-trial basis when an odor was repeatedly presented (blue line, *Figure 4B*). In the end, the weight S1 may be enhanced while the weight S2 may be suppressed. Although odor representations of M/T cells remained stable with and without STDP (*Figure 4C1*), we found that odor representations in PCx were stabilized with STDP (*Figure 4C2*). The correlation drops from Day-0 were significantly reduced (0.8 without STDP and 0.4 with STDP by Day-10, *Figure 4C2*, left), and the drift rate (° per day), which quantified the daily angle changes of the population responses (see **Methods**), was reduced by 26% (7.9 without STDP and 5.8 with STDP, *Figure 4C2*, right). Adding STDP to the network and using repeated presentations of an odor meant that the synapses that would otherwise randomly change due to adult-neurogenesis converged asymptotically to some weight (*Figure 4—figure supplement 1*). Importantly, we identified that while the brain is constantly undergoing change, especially in the MOB and the PCx due to ongoing adult-neurogenesis, the impact of these changes on both population responses and the representation depends as m uch on the plasticity rules as it does on the ongoing statistics of the environment.

## Discussion

In our study, we modeled adult-neurogenesis as a dynamic reshuffling of GCs within a detailed spiking neuronal network that replicated the circuit architecture of the MOB and PCx. This modeling revealed how adult-neurogenesis differentially modulates odor responses and representations in M/T cells and PCs.

In the MOB, individual M/T cells exhibited variable odor responses with changing firing rate magnitudes over time. This is consistent with earlier experimental studies using calcium-imaging (*Kato et al., 2012*; *Yamada et al., 2017*; *Shani-Narkiss et al., 2023*). Variability in M/T cell responses arises on multiple time scales. Experimental studies show pronounced trial-to-trial variability within single recording sessions (*Wesson et al., 2008*; *Padmanabhan and Urban, 2010*; *Angelo and Margrie, 2011*; *Kapoor and Urban, 2006*; *Friedrich and Laurent, 2001*; *Smear et al., 2011*), which cannot be attributed to adult-neurogenesis. This fast variability likely reflects ongoing network dynamics (*Laurent, 2002*), behavioral state (*Kato et al., 2012*; *Schreck et al., 2022*; *Chockanathan et al., 2021*), attention (*D'Souza and Vijayaraghavan, 2014*), or top-down modulation (*Chen and Padmanabhan, 2022*; *Boyd et al., 2012*; *Otazu et al., 2015*), none of which are included in our model. Our model therefore captures only a limited amount of within-session variability and instead focuses on slower changes that accumulate across days. In this regime, adult-neurogenesis provides a mechanism for gradual changes in individual M/T responses. It would be interesting for future work to integrate both fast and slow sources of variability within a unified modeling framework.

Despite these fluctuations in individual cells, the overall pattern of M/T cells population responses remained stable. This stability is attributed to consistent glomerular input, as reflected in the low-dimensional M/T cell trajectories and their proximity in reduced PCA space. This stability of M/T cells we observed in the reduced space is consistent with a recent experimental study (*Shani-Narkiss et al., 2023*) using both linear (PCA) and nonlinear (t-SNE) dimensionality reduction methods. Prior experimental studies report heterogeneous effects of experience on mitral cell odor responses. Repeated

odor exposure has been associated with increased sparsening and reduced response amplitude in some experiments (*Kato et al., 2012*), whereas others report substantial reorganization of ensemble representations across days without consistent sparsening (*Yamada et al., 2017*). Our model best recapitulates the findings of the latter, showing that adult-neurogenesis can reorganize individual M/T responses while preserving stable low-dimensional population structure. Differences in behavioral features across studies may reflect variations in behavioral context or top-down modulation, factors that are not explicitly modeled here.

Conversely, in the PCx, both individual and population patterns of PCs responses were in constant flux due to the ongoing adult-neurogenesis. This resulted in a geometric reshaping of odor representations and a drift in the odor manifold defined by those representations, consistent with experimental findings (*Schoonover et al., 2021*). Our results reveal how the process of adult-neurogenesis may support a number of computations in the early olfactory circuits, including how a sensory representation may remain stable, and how it might change due to plasticity.

We note that our simulations do not assume complete replacement of the GC population over the 11-day period. Rather, each simulated 'day' represents a discrete epoch used to implement plasticity across multiple time scales. While some developmentally born GCs can persist for much of the animal's lifetime (*Magavi et al., 2005*), some fraction of the population undergoes turnover through adult-neurogenesis (*Lepousez et al., 2013*; *Kaplan et al., 1985*; *Petreanu and Alvarez-Buylla, 2002*). In our model, turnover is applied uniformly across GCs, without preferential survival. Additionally, different simulated time points may also be thought of as reflecting regimes ranging from higher turnover to partial preservation of existing GCs. While behavioral factors such as novelty or running can modulate neurogenesis rates (*Kamimura et al., 2022*; *van Praag et al., 1999*; *Gheusi and Lledo, 2014*), these effects are not included in our study and would be important for future studies.

Second, we found that one of the roles for STDP in the early olfactory system is reducing representational drift in the PCx. Repeated exposure to an odor significantly reduced the representational drift in the PCx, consistent with experimental observations (*Schoonover et al., 2021*). Rather counterintuitively, this result suggests that a role of STDP and plasticity in general is stabilizing representations despite ongoing changes in the circuit. This implicates STDP at abGC-related synapses in a broader functional context. Previous experimental studies have shown that abGCs integrate into the olfactory bulb in an activity-dependent manner and preferentially stabilize circuits engaged by behaviorally relevant odors (*Livneh and Mizrahi, 2012*; *Wu et al., 2020*; *Alonso et al., 2006*). Our model is consistent with this view, but approaches the problem at the level of synaptic plasticity rather than cell survival. We show that STDP acting on abGC synapses can reduce representational drift by selectively stabilizing dimensions of population activity that are repeatedly activated due to experience. Here, representational stability arises because of the stability of the sensory world; representational permanence, rather than being a feature of the circuit, is a feature of the sensory landscape, a stabilizing force against the inherent variability introduced by continuous adult-neurogenesis. Adult-neurogenesis, which occurs throughout the animal's life, may confer a mechanistically different computational framework as compared to vision or audition, for which critical periods delineate the bounds of plasticity and define the periods over which sensory representations are changed or stabilized.

Third, our various dimensionality reduction analyses aim to illustrate the structure of population activity in terms of the variance and covariance of neuronal responses; which features of this activity are most relevant likely depend on what downstream brain regions receive this input and what computations these regions ultimately perform. In this framework, both low and high-dimensional representations would be functionally relevant but may serve different roles. In the bulb, the low-dimensional population structure remains stable despite ongoing reorganization in the full ensemble, consistent with its role in providing a reliable sensory code. In contrast, PCx exhibits drift even in its low-dimensional space, reflecting its more flexible and higher-dimensional representational geometry. This higher dimensionality arises from divergent bulb inputs, dense recurrent connectivity, and integration of contextual and feedback signals. Importantly, STDP reduces drift in PCx by stabilizing the dimensions reinforced by experience. Residual drift persists because the cortical population retains many orthogonal dimensions through which representations can continue to evolve.

Fourth, our model assumes broadly distributed projections from MOB to PCx and random intra-piriform connectivity (*Stern et al., 2018*; *Haberly and Bower, 1984*; *Franks et al., 2011*). These divergent projections from MOB to PCx are central to the combinatorial coding strategy of PCx,

allowing individual pyramidal neurons to represent complex mixtures and higher-order odor features (*Bekkers and Suzuki, 2013*; *Apicella et al., 2010*; *Blazing and Franks, 2020*). On the other hand, recent studies have revealed increasing structure in olfactory connectivity, including molecularly defined projection neuron subtypes, long-range functional loops, and experience-dependent motifs within PCx (*Chae et al., 2022*; *Zeppilli et al., 2021*; *Fink et al., 2025*). These findings point to additional constraints that may shape learning and perception. An important direction for future work will be to incorporate such structured connectivity and cortical plasticity into our model to examine how they interact with adult-neurogenesis in the bulb to influence representational drift in PCx.

In addition to PCx, M/T cells also project to other cortical areas such as the anterior olfactory nucleus (AON). Although our model does not differentiate between mitral and tufted cells, it has been recently reported that mitral and tufted cells have distinct preferred cortical targets, with mitral cells preferentially projecting to PCx and tufted cells preferentially projecting to AON (*Chae et al., 2022*). Tufted cells substantially outperform mitral cells in decoding both odor identity and intensity, suggesting that tufted cells and the AON to which they project are ideal for stable odor identity compared to the mitral cells and PCx (*Chae et al., 2022*). As mitral cells are located in the deeper layers of the bulb, they interact with abGCs, whereas tufted cells in the superficial layer may preferentially interact with existing GCs that were born in the neonatal period (*Lemasson et al., 2005*; *Whitman and Greer, 2009*). Therefore, the activity of mitral cells may be more impacted by the turnovers of abGCs, and the effect of adult-neurogenesis percolates onto PCx. By contrast, adult-neurogenesis has less impact on the tufted cells and the AON, potentially contributing to the stable encoding of odors in AON. This is one prediction of our model.

While our study focused on the effects of adult-neurogenesis in the olfactory bulb on odor representations, it does not negate the possibility of other forms of plasticity (*Qin et al., 2023*) contributing to the representational drift observed in the PCx (*Schoonover et al., 2021*). Indeed, similar representational drifts have been noted in other brain areas like the posterior parietal cortex (PPC) (*Driscoll et al., 2017*), primary motor cortex (*Rokni et al., 2007*), and hippocampus (*Lee et al., 2020*; *Gonzalez et al., 2019*). Of these regions, adult-neurogenesis only occurs in the hippocampus, where adult-born GCs integrate into the existing circuitry of dentate gyrus (*Toda et al., 2019*). Interestingly, similar to what we found in the early olfactory system (i.e., newborn cells in MOB and drift in PCx), the cells downstream of the dentate in the CA3 region (*Hainmueller and Bartos, 2018*) and CA1 region (*Ziv et al., 2013*; *Lee et al., 2020*; *Gonzalez et al., 2019*) exhibit drift over days and weeks.

Both PCx and hippocampus are evolutionarily old archi-cortical structures. This conservation of adult-neurogenesis in these brain regions may suggest a broader evolutionary strategy for balancing stability and flexibility. First, adult-neurogenesis in the olfactory bulb may support ongoing plasticity due to continuous peripheral turnover of olfactory receptor neurons (*Lepousez et al., 2013*; *Lazarini and Lledo, 2011*). In this context, abGCs may provide a mechanism for adapting bulb circuitry to changing input statistics, injury, or learning demands. Second, adult-neurogenesis may confer selective advantages for navigating dynamic environments. By ensuring flexibility in neural circuits, such as through adult-neurogenesis, these regions can rapidly update the representations of odors and spatial information, respectively, in response to changing environmental cues. This flexibility may enable the early organisms to effectively represent and adapt to the dynamic worlds they inhabited, enhancing their chances of survival and reproduction across evolutionary time scales (*Gonçalves et al., 2016*; *Jurkowski et al., 2020*; *Sailor et al., 2017*). As organisms evolved, evolutionarily newer brain regions such as the PPC, lacking significant adult-neurogenesis, developed alternative mechanisms to balance plasticity and stability (*Qin et al., 2023*). Our work thus offers a new perspective on how the brain adapts to an ever-changing sensory environment.

## Methods
### Organization and architecture of the model
The MOB consisted of 50 glomeruli (G) corresponding to the olfactory receptor neuron inputs into the MOB (*Mombaerts et al., 1996*). Each glomerulus was connected to 25 M/T cells for a total 1250 M/T cells. Within the MOB, a local population of 12,500 inhibitory GCs formed reciprocal and lateral inhibitory connections with M/T cells. Individual M/T cell 'projections' formed random excitatory connections with 10,000 piriform cortical cells (PCs) in PCx. These PCs in turn 'projected' back to the olfactory

bulb, providing excitatory centrifugal feedback onto the inhibitory GCs in the bulb. Within PCx, two types of inhibitory interneurons were included: a population of 1250 feedforward inhibitory neurons (FFIs) that received excitatory input from M/T cells and inhibited both PCs and other FFIs, and a population of 1250 local feedback inhibitory neurons (FBIs) that received input from a random subset of PCs and subsequently inhibited PCs and other FBIs.

## Voltage dynamics of individual neurons

The voltage dynamics of individual cells in the network are modeled as spiking neurons (*Izhikevich, 2003*) described by a two-dimensional system of ordinary differential equations of the form,

$$\frac{dv}{dt} = 0.04v^2 + 5v + 140 - u + I$$
$$\frac{du}{dt} = a\left(bv - u\right) \qquad (1)$$

with the after-spiking resetting

$$\text{if } v \geq 30mV, \text{ then } v \leftarrow c, \, u \leftarrow u + d. \qquad (2)$$

Here, $v$ represents the voltage (mV) of the neuron and $u$ represents a dimensionless membrane recovery variable accounting for the activation or inactivation of ionic currents; $t$ is time and has unit of ms; $a$, $b$, $c$, and $d$ are the parameters by tuning which various firing patterns can be generated; $I$ represents synaptic currents or injected DC currents to the neuron.

We choose to use this neuron model to simulate the voltage dynamics of individual neurons because: (1) It combines the biological plausibility of the Hodgkin–Huxley neuron model and the computational efficiency of leaky integrate-and-fire neuron model, allowing us to simulate tens of thousands of spiking neurons simultaneously in our network; (2) Different combinations of the parameter values $a$, $b$, $c$, and $d$ can reproduce a diversity of firing patterns of neurons of known types, so we can capture the biophysical diversity in the firing properties for different types of neurons in the olfactory system, such as the M/T cells and GCs in the MOB, and piriform cortical cells and other local inhibitory interneurons in PCx. In order to achieve heterogeneity such that different cells within the same type exhibit different dynamics, we introduced randomness in the parameter assignment (see *Table 1*). The $r_i$ is a random variable uniformly distributed on the interval $[0, 1]$ and $i$ denotes the neuron index. For example, the parameter $a$ will be distributed on the interval $[0.02, 0.1]$ within which various firing patterns can emerge. We also used $r_i^2$ or $r_i^4$ to bias the distribution to different extents for different cell types. Overall, based on our choice of parameters in the Izhikevich model, the spiking patterns of PCs and M/T cells largely fall into the category of regular spiking, intrinsically bursting, or chattering neurons (*Davison and Ehlers, 2011*; *Connors and Gutnick, 1990*; *Padmanabhan and Urban, 2014*). Inhibitory neurons including GCs and FFIs/FBIs in the network can generate spiking patterns as fast spiking neurons and low-threshold spiking neurons (*Burton and Urban, 2015*; *Egger et al., 2005*; *Gibson et al., 1999*; *Suzuki and Bekkers, 2012*).

Synaptic input $I$ to each neuron depends on the neuron type. For a cell $i$ in MOB, $I_i$ is a linear superposition of various sources

$$I_i = I_i^{mc-ex} + I_i^{gc-in} + I_i^{osn} + I_i^{feedback} + \xi_i. \qquad (3)$$

Here, $I_i^{mc-ex}$ represents excitation from M/T cells and exists for both M/T cells and GCs. For GCs, when a M/T cell fires, the excitatory post-synaptic current $I_i^{mc-ex}$ into different GCs are delayed by

**Table 1.** Parameters of Izhikevich neuron model for different cell types.

|   | M/T cells | GCs | PCs | FFIs/FBIs |
|---|---|---|---|---|
| $a$ | $0.1 - 0.08r_i^4$ | $0.1 - 0.08r_i^2$ | $0.02 + 0.08r_i$ | $0.1 - 0.08r_i^2$ |
| $b$ | $0.2$ | $0.2$ | $0.2$ | $0.2$ |
| $c$ | $-65$ | $-65 + 15r_i^2$ | $-65$ | $-65 + 15r_i^2$ |
| $d$ | $2 + 6r_i^4$ | $2$ | $8 - 6r_i$ | $2$ |

different latencies, resulting in different spiking latencies of GCs, consistent with previous experimental findings in the olfactory bulb GC network (**Kapoor and Urban, 2006**). The $I_i^{gc-in}$ represents inhibition from GCs and exists for both M/T cells and GCs. $I_i^{osn}$ represents glomerular input and only exists for M/T cells. When a glomerulus is activated by a model odor, it provides correlated inputs $I_i^{osn}$ to the M/T cells driven by that glomerulus. When a glomerulus is activated, the input current $I_i^{osn}$ that an associated M/T cell receives is modeled as a step function with Gaussian noise added. Since each glomerulus receives inputs from a set of receptor neurons expressing the same olfactory gene type, the inputs to individual glomeruli from receptor neurons are substantially correlated (**Dhawale et al., 2010**; **Koulakov and Rinberg, 2011**; **Lledo et al., 2005**; **Wachowiak et al., 2004**). Therefore, we assumed that the glomerular inputs to the apical dendrites received by the M/T cells associated with the same glomerulus were correlated. No input correlation between M/T cells associated with different glomeruli is assumed. $I_i^{feedback}$ represents excitatory centrifugal input from PCs and is non-zero only for GCs when feedback is ON. We set it to zero for all GCs when feedback is OFF. The $\xi_i$ represents Gaussian white noise input with zero mean and standard deviation $\sigma = 1.75$ for M/T cells and $\sigma = 0.8$ for GCs.

Similarly, for a cell $i$ in PCx, $I_i$ is composed of

$$I_i = I_i^{mob} + I_i^{pc-ex} + I_i^{in} + \eta_i, \tag{4}$$

where $I_i^{mob}$ represents input from M/T cells in MOB and only exists for piriform cortical cells (PCs) and FFIs; $I_i^{pc-ex}$ represents excitation from PCs and exists for both PCs and FBIs; $I_i^{in}$ represents inhibition from local inhibitory neurons including FFIs and FBIs; $\eta_i$ represents Gaussian white noise input (zero mean and standard deviation $\sigma = 0.9$) and only exists for PCs.

Each action potential fired by a presynaptic neuron will evoke a jump in the corresponding synaptic inputs of all its postsynaptic targets by an amount equal to the appropriate synaptic strength. For example, action potentials of a M/T cell induce jumps in the excitatory currents of their postsynaptic target neurons, including $I_i^{mc-ex}$ in M/T cells and GCs in MOB, and $I_i^{mob}$ in FFIs and PCs in PCx. These synaptic inputs then decay to zero with time constant 10 ms. The height of the jump is determined

**Table 2.** Network parameters controlling the connectivity between cell types.

| | Connection density | Average synaptic strength | References |
|---|---|---|---|
| MCtoMC (same glomerulus) | 0.8 | 0.25 | |
| MCtoMC (different glomeruli)* | 0 | 0 | **Urban and Sakmann, 2002**; **Egger and Urban, 2006**; **Schoppa and Westbrook, 2002**; **Soucy et al., 2009** |
| MC2GC | 0.3 | 0.25 | **Schoppa and Urban, 2003**; **Kapoor and Urban, 2006**; **Burton and Urban, 2015**; **Willhite et al., 2006** |
| GC2MC | 0.02 | −0.4 | |
| GC2GC | 0.05 | −0.1 | **Boyd et al., 2012**; **Burton and Urban, 2015** |
| MC2PC | 0.5 | 0.06 | **Sosulski et al., 2011**; **Miyamichi et al., 2011**; **Stern et al., 2018**; **Luna and Schoppa, 2008** |
| MC2FFI | 0.2 | 0.2 | |
| PC2GC | 0.9 | 0.03 | **Otazu et al., 2015**; **Nunez-Parra et al., 2013**; **Padmanabhan et al., 2018** |
| PC2PC | 0.01 | 0.1 | |
| FFI2PC | 0.1 | −0.1 | |
| FBI2PC | 0.8 | −0.1 | |
| FFI2FFI | 0.01 | −1.0 | |
| PC2FBI | 0.02 | 0.3 | **Stern et al., 2018**; **Franks et al., 2011**; **Suzuki and Bekkers, 2012**; **Large et al., 2016** |
| FBI2FBI | 0.02 | −0.5 | |

*We assumed zero connection between M/T cells associated with different glomeruli.

by the pairwise synaptic strength between any two neurons and their values are given in the synaptic weight matrix which will be described in the next section.

## Synaptic strength and model network architecture

The MOB consists of 50 glomeruli, each of which drives 25 M/T cells, thus a total of 1250 M/T cells in MOB. Besides, a local population of 12,500 inhibitory GCs formed reciprocal and lateral inhibitory connections with M/T cells. Thus, within the MOB, we have a weight matrix $\mathbf{W}_{mob}$ of 13,750 by 13,750 with its entry $W_{mob}^{ij}$ representing the synaptic strength from presynaptic neuron $j$ to postsynaptic neuron $i$. Depending on the cell type, this matrix $\mathbf{W}_{mob}$ can be partitioned into four sub-matrices, i.e., from M/T cell to M/T cell, from M/T cell to GC, from GC to M/T cell, and from GC to GC. The specific value of each entry in $\mathbf{W}_{mob}$ was assigned randomly according to two parameters we chose for each sub-matrix. One is the connection density (the percentage of non-zero synaptic weights) and the other is the average synaptic strength (mean of a uniform distribution from which individual synaptic weights are sampled). Each sub-matrix has its own value of the connection density and average synaptic strength. In particular, the connection density and average synaptic strength between M/T cells driven by the same glomerulus are higher than between M/T cells driven by different glomeruli.

Individual M/T cell 'projections' form random excitatory connections with 10,000 PCs and 1250 FFIs in PCx, giving rise to a feedforward weight matrix $\mathbf{W}_{ff}$ of 11,250 by 1250. Within PCx, PCs form recurrent excitations with each other. The FFIs inhibit both PCs and other FFIs, and another population of 1250 FBIs that receive input from a random subset of PCs inhibit PCs and other FBIs. Therefore, we have a matrix $\mathbf{W}_{pcx}$ of 12,500 by 12,500 that identifies all synaptic weights between cells in PCx. PCs 'project' back to the MOB, providing excitatory centrifugal feedback to GCs, giving rise to a feedback weight matrix $\mathbf{W}_{fb}$ of 12,500 by 10,000. Under the condition of centrifugal feedback OFF, this $\mathbf{W}_{fb}$ is set to be a zero matrix. The connection density and average synaptic strength for all sub-matrices can be found in *Table 2*. The parameters are all chosen heuristically based on previous theoretical and experimental studies listed in *Table 2*.

Feedback projections from PCx to the bulb may be structured, as retrograde rabies tracing demonstrates that PCs projecting to GC populations in the bulb tend to be spatially clustered (*Padmanabhan et al., 2016*). Furthermore, a number of studies suggest that GC synapses are especially sensitive to plasticity (*Livneh and Mizrahi, 2012*; *Sailor et al., 2016*), either through adult-neurogenesis or more traditional mechanisms of synaptic reorganization. To implement all of these features, we structure the feedback projections to GCs such that the PCs receiving feedforward inputs from the M/T cells of certain glomeruli project back to the GCs which are reciprocally connected with M/T cells associated with other glomeruli. Reciprocal connectivity between M/T cells and GCs is defined as: M/T-1 excites GC-1 and GC-1 inhibits M/T-1, as observed by many studies (*Wanner and Friedrich, 2020*; *Willhite et al., 2006*). Across the M/T population, there are 291 ± 9 (mean ± SD, $n$ = 1250 M/T cells) GCs that are reciprocally connected with each M/T cell. As a result, each PC projects to 7368 ± 64 GCs (mean ± SD, $n$ = 10,000 PCs) with weight magnitude larger than 0.01. All feedback synaptic weights are randomly generated with small magnitude less than 0.05, and this structure gives rise to a dense but weak connectivity matrix $\mathbf{W}_{fb}$. Due to the sparsity of the PC firings when feedback is ON, this dense and weak top-down connectivity ensures robust influence of PCs on GC activity and thus the contribution of PCx on odor processing in MOB.

## Model odor definition

Model odors are defined by the combinatorial patterns of glomeruli which are activated successively with different glomerular timing, a pattern recapitulating the spatiotemporal structure of odor inputs (*Rubin and Katz, 1999*; ; *Meister and Bonhoeffer, 2001*). Specifically, when a model odor is presented, 3–10 glomeruli will be activated (6–20% of all glomeruli) and all the M/T cells associated with those glomeruli will receive correlated glomerular input $I^{osn}$ which lasts for 90 ms. A table of 100 model odors was defined as the odor inputs to our network.

## Modeling adult-neurogenesis as weight reshuffling

Adult-neurogenesis of the GCs constantly removes old GCs and replaces them with adult-born ones. As a consequence, all the synapses from and onto old GCs are removed and new synapses with adult-born ones are built. We modeled this process by weight reshuffling in the network with the

total number of GCs fixed. On each simulation day, 10% of the GCs had their synaptic weights reshuffled. For each GC, the values of synaptic weights changed randomly on each day. The distributions from which new synaptic weights were sampled were the same distributions as building the network (see *Table 2*). We only reshuffled the individual weight values without changing the whole weight distributions.

### Principal component analysis

Spiking activity of each M/T cell and each PC was binned into a 5-ms sliding time window and averaged across trials (each model odor was presented in 10 trials). To perform the PCA analysis, we concatenated the trial-averaged responses of all M/T cells to all 100 model odors on all simulation days, resulting in a large matrix of 1250 cells by 247 time bins × 100 odors × 11 days. Response covariance matrices (1250 by 1250) were computed for this concatenated matrix (after subtracting the mean responses averaged across time bins, odors, and days). This gave us a single set of eigenvectors, thus the same eigenspace into which cell responses for all days can be projected and compared. Each 1250-dimensional cell response vector was then projected onto the first 3 principal eigenvectors for visualization and the first 50 principal eigenvectors for computations. The same procedure was also done for PCs. In our simulations, these PCs captured the majority of variance relevant for odor identity (~60–70% for M/T cells and ~55–65% for PCx).

### Population vectors of firing rates and PCA trajectories

We constructed the population vectors using either the firing rates of all cells or the PCA trajectories of the first 50 dimensions. When using the firing rates, for each odor on each day, the single-trial responses of all the cells was a matrix of 1250 cells by 247 time bins for M/T cells or 10,000 cells by 247 time bins for PCs. We then converted the matrix into a long vector of lengths 1250 for M/T cells and 10,000 for PCs. When using PCA trajectories, the same procedures were applied, with only the cell number replaced by the reduced dimensionality.

### Pairwise correlation between population vectors

To measure the similarity of population responses over time, we computed the Pearson's pairwise correlations of the population vectors constructed either by firing rates or PCA trajectories on two comparison days (e.g., day-i and day-j). For within-odor correlation, the two population vectors were the responses (either single-trial or trial-averaged) to the same odor on day-i and day-j. For across-odor correlation, the two population vectors were the responses to any given two different odors on day-i and day-j, and we computed that for all different odor pairs on those 2 days. When $i = j$, the within-odor correlation was computed by comparing responses of even and odd trials.

### Cosine similarity between population vectors

To gain a geometric perspective of the drift over days, we computed the cosine similarity of the population vectors by $\theta_{i,j} = \boldsymbol{u}_i \cdot \boldsymbol{u}_j / \|\boldsymbol{u}_i\| \times \|\boldsymbol{u}_j\|$, where $\theta_{i,j}$ is the cosine similarity between day-i and day-j, and $\boldsymbol{u}_i$ ($\boldsymbol{u}_j$) is the trial-averaged population vector on day-i (day-j). We also estimated the within-day variability for each odor on each day by computing the cosine similarity between the even trial-averaged and odd trial-averaged responses. For within-day cosine similarity when $i = j$, we subtracted the estimated within-day variability from the computed cosine similarity.

### Decoding analysis: K nearest neighbor algorithm

The K nearest neighbor approach was used to decode odor identity from the projected ensemble responses of PCs to any given odor pair (*Padmanabhan and Urban, 2010*). Consistent with the computation of symmetrized Kullback–Leibler divergence $D_{KL}$, analysis was performed in the space of the first 50 principal components. The original data was broken up into testing and training sets. The training sets established the location of PC responses to known odors (i.e., known PC responses) in the principal component space and the testing sets were probed with respect to these known PC responses. The Euclidean distance of the unknown odors to all PC responses was then calculated and the K nearest neighbors were used to determine to which odor the unknown PC activity was responding to. This process of generating testing and training sets was repeated 30 times, with each repeat reflecting a different random population of testing and training to ensure that the decoding

accuracy was not a result of artifacts of selecting a single testing/training population. Free parameters in the K nearest neighbor algorithm include the ratio of testing to training data, and the number of nearest neighbors used in the calculation. For training/testing, we used ratios of 50%, 70% and 90%. We also examined the algorithm's accuracy when 3, 5, and 7 nearest neighbors were used.

### Spike-timing-dependent plasticity

To model the effect of spike trains on synaptic weights, we used the suppression model given in *Froemke and Dan, 2002*. Each pre- and post-synaptic spike was assigned an efficacy $\epsilon_i = 1 - e^{-(t_i - t_{i-1})/\tau_s}$, where $\epsilon_i$ is the efficacy of the $i$th spike, $t_i$ and $t_{i-1}$ are the timing of the $i$th and $(i-1)$th spike, respectively, and $\tau_s$ is the suppression time constant. The effect of each pair of pre- and post-synaptic spikes on synaptic modification was given by $\Delta w_{ij} = \epsilon_i^{pre} \epsilon_j^{pos} F(\Delta t_{ij})$, where $\epsilon_i^{pre}$ and $\epsilon_j^{pos}$ are the efficacies of the $i$th presynaptic spike and $j$th postsynaptic spike, respectively, and $\Delta t_{ij}$ is the interval between the two spikes $t_j^{pos} - t_i^{pre}$. The function $F$ represents the effect of the temporal window for STDP, defined as:

$$F(\Delta t) \begin{cases} A_+ e^{-|\Delta t|/\tau_+} & if \quad \Delta t > 0 \\ A_- e^{-|\Delta t|/\tau_-} & if \quad \Delta t < 0 \end{cases},\tag{5}$$

where $A$ is the scaling factor, $\tau$ is the time constant, $+$ means LTP, and $-$ means long-term depression. We chose $\tau_s^{pre} = 34$ms, $\tau_s^{pos} = 75$ms. $A_{+/-}$ and $\tau_{+/-}$ are drawn from normal distributions for each synapse, where $A_+ \sim \mathcal{N}(1.03, 0.1)$, $A_- \sim \mathcal{N}(-0.51, 0.01)$, $\tau_+ \sim \mathcal{N}(13.3, 1.7)$ (in ms), $\tau_- \sim \mathcal{N}(34.5, 1.6)$ (in ms), and $\mathcal{N}(\mu, \sigma)$ represent normal distribution with mean $\mu$ and standard deviation $\sigma$. In our model, STDP acts on two sets of connections. It applies to the synapses *onto* abGCs from M/T cells, GC/SAC cells, and PCx neurons. It also applies to the synapses *from* abGCs, including those onto M/T cells and GC/SAC cells.

## Acknowledgements

This study was supported by funding from the National Institutes of Health (NIH) and the National Science Foundation (NSF). KP was funded by NIH R01 MH113924, NSF CAREER 1749772, the Cystinosis Research Foundation, and the Kilian J and Caroline F Schmitt Foundation. This manuscript has been released as a preprint.

## Additional information

### Funding

| Funder | Grant reference number | Author |
|---|---|---|
| National Institute of Neurological Disorders and Stroke | NS135763 | Krishnan Padmanabhan |
| National Institute on Deafness and Other Communication Disorders | DC021141 | Krishnan Padmanabhan |
| National Institute of Mental Health | MH113924 | Krishnan Padmanabhan |
| U.S. National Science Foundation | 1749772 | Krishnan Padmanabhan |
| Cystinosis Research Foundation | | Krishnan Padmanabhan |
| Kilian J and Caroline F Schmitt Foundation | | Krishnan Padmanabhan |

The funders had no role in study design, data collection, and interpretation, or the decision to submit the work for publication.

### Author contributions
Zhen Chen, Software, Formal analysis, Validation, Investigation, Visualization, Writing - original draft, Writing – review and editing; Krishnan Padmanabhan, Conceptualization, Formal analysis, Supervision, Funding acquisition, Investigation, Visualization, Project administration, Writing – review and editing

### Author ORCIDs
Zhen Chen ⬥ https://orcid.org/0000-0002-5590-3552
Krishnan Padmanabhan ⬥ https://orcid.org/0000-0002-3255-8346

Reviewer #1 (Public review): https://doi.org/10.7554/eLife.107905.3.sa1
Reviewer #3 (Public review): https://doi.org/10.7554/eLife.107905.3.sa2
Author response https://doi.org/10.7554/eLife.107905.3.sa3

## Additional files

### Supplementary files
MDAR checklist

### Data availability
The current manuscript is a computational study, so no data have been generated for this manuscript. All source code is provided in GitHub (copy archived at *Chen, 2026*).

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
