## [Editor Report · eLife Assessment]

This paper presents an **important** theory and analysis of the role of neurogenesis and inhibitory plasticity in the drift of neural representations in the olfactory system. For one of the findings, regarding the impact of neurogenesis on the drift, the evidence remains **incomplete**. The reason lies in the differences in variability/drift of the mitral/tufted cell responses observed in the model compared to experimental observations, where these responses remain stable over extended time scales.

---

## [Referee Report · Reviewer #1 (Public review)]

Summary:

The authors build a network model of the olfactory bulb and the piriform cortex and use it to run simulations and test their hypotheses. Given the the model's settings, the authors observe drift across days in the responses to the same odors of both the mitral/tufted cells, as well as of piriform cortex neurons. When representing the M/T and PCx responses within a lower dimensional space, the apparent drift is more prominent in the PCx, while the M/T responses appear in comparison more stable. The authors further note that introducing spike-time dependent plasticity (STDP) at bulb synapses involving abGCs slows down the drift in the PCx representations, and further link this to the observation that repeated exposure to the same odorant slows down drift in the piriform cortex.

The model is clearly explained and relies on several assumptions and observations: (1) random projections of MTC from the olfactory bulb to the piriform cortex, random intra-piriform connectivity and random piriform to bulb connectivity; (3) higher dimensionality of piriform cortex representations compared to M/T responses which enables superior decoding of odor identity in the piriform cortex; (2) spike time dependent plasticity (STDP) at synapses involving the abGCs.

The authors address an open topical problem and model is elegant in its simplicity. The authors addressed many of my concerns by plotting new analyses and by adding clarifying statements and discussion points, as well as testable predictions to the revised manuscript. In the revised manuscript, a few points remain unclear and I am listing them below for further potential discussion.

(1) Given the large in response (variability) across trials reported by Shani-Narkiss, Kay & Laurent - the question remains open: what fraction of the variability in response across days can be really accounted by adult born neurogenesis (the main topic of this study) vs. other mechanisms. I think the answer to this question is key for interpreting the results presented by the authors on the impact of adult neurogenesis on changes of mitral cell responses. Unfortunately, I could not find the answer in the revised version of the manuscript.

(2) Yamada indeed reported a "drastic reorganization of ensemble odor representation" in their manuscript (Figure 3D), but my understanding is that this was observed in the context of passive exposure to the same odor across several days in a row. This does not appear to contradict the findings of Kato et al., 2012 that when an odor is presented seldom, across days the mitral cell responses are stable. Also, data from Yamada et al. appears to show some degree of overall sparsening of odor responses in mitral cells at least at the level of a decrease in response amplitude between day 1 to day 7 of repeated passive exposure (Figure 3A, Yamada et al., 2017).

(3) There was mistake on my part on one of the papers referenced with respect to random vs. structured projections from the olfactory bulb to the piriform cortex. The one I was referring to is Chen et al., Cell, 2022 (not Chae et al., Neuron, 2022). The authors discussed the implications from the latter, while I was commenting in fact on the findings from Chen et al., 2022. This study identified structured projections of individual mitral cells along the A-P axis of the piriform cortex in conjunction with collaterals to specific subsets of extra-piriform target regions.

---

## [Referee Report · Reviewer #3 (Public review)]

Summary

The authors set out to explore the potential relationship between adult neurogenesis of inhibitory granule cells in the olfactory bulb and cumulative changes over days in odor-evoked spiking activity (representational drift) in the olfactory stream. They developed a richly detailed spiking neuronal network model based on Izhikevich (2003), allowing them to capture the diversity of spiking behaviors of multiple neuron types within the olfactory system. This model recapitulates the circuit organization of both the main olfactory bulb (MOB) and the piriform cortex (PCx), including connections between the two (both feedforward and corticofugal). Adult neurogenesis was captured by shuffling the weights of the model's granule cells, preserving the distribution of synaptic weights. Shuffling of granule cell connectivity resulted in cumulative changes in stimulus-evoked spiking of the model's M/T cells. Individual M/T cell tuning changed with time, and ensemble correlations dropped sharply over the temporal interval examined (long enough that almost all granule cells in the model had shuffled their weights). Interestingly, these changes in responsiveness did not disrupt low-dimensional stability of olfactory representations: when projected into a low-dimensional subspace, population vector correlations in this subspace remained elevated across the temporal interval examined. Importantly, in the model's downstream piriform layer this was not the case. There, shuffled GC connectivity in the bulb resulted in a complete shift in piriform odor coding, including for low-dimensional projections. This is in contrast to what the model exhibited in the M/T input layer. Interestingly, these changes in PCx extended to the geometrical structure of the odor representations themselves. Finally, the authors examined the effect of experience on representational drift. Using an STDP rule, they allowed the inputs to and outputs from adult-born granule cells to change during repeated presentations of the same odor. This stabilized stimulus-evoked activity in the model's piriform layer.

Strengths

This paper suggests a link between adult neurogenesis in the olfactory bulb and representational drift in the piriform cortex. Using an elegant spiking network that faithfully recapitulates the basic physiological properties of the olfactory stream, the authors tackle a question of longstanding interest in a creative and interesting manner. As a purely theoretical study of drift, this paper presents important insights: synaptic turnover of recurrent inhibitory input can destabilize stimulus-evoked activity, but only to a degree, as representations in the bulb (the model's recurrent input layer) retain their basic geometrical form. However, this destabilized input results in profound drift in the model's second (piriform) layer, where both the tuning of individual neurons and the layer's overall functional geometry are restructured. This is a useful and important idea in the drift field and to my knowledge is novel. The bulb is not the only setting where inhibitory synapses exhibit turnover (whether through neurogenesis or synaptic dynamics), and so this exploration of the consequences of such plasticity on drift is valuable. The authors also elegantly explore a potential mechanism to stabilize representations through experience, using an STDP rule specific to the inhibitory neurons in the input layer. This has an interesting parallel with other recent theoretical work on drift in the piriform (Morales et al., 2025 PNAS), in which STDP in the piriform layer was also shown to stabilize stimulus representations there. It is fascinating to see that this same rule also stabilizes piriform representations when implemented in the bulb's granule cells.

The authors also provide a thoughtful discussion regarding differential roles of mitral and tufted cells in drift in piriform and AON and potential roles of neurogenesis in archicortex.

In general, this paper puts an important and much-needed spotlight on the role of neurogenesis and inhibitory plasticity in drift. In this light, it is a valuable and exciting contribution to the drift conversation.

Comments on revisions:

I appreciate the substantial revisions the authors have made to the manuscript. The paper is clearly improved and addresses an important and timely question: the relationship between adult neurogenesis and drift. In particular, the effort to link adult neurogenesis in the olfactory bulb to the long-term stability of odor representations downstream is valuable, and the modeling provides useful mechanistic intuition about how inhibitory circuit remodeling could influence representational drift across layers.

That said, I remain concerned that the manuscript, as currently framed, risks giving readers the incorrect impression that experimental work has established progressive, time-dependent drift in the odor tuning of olfactory bulb neurons. Experimental studies do show that ongoing experience with a set of odors can profoundly alter bulbar responses to those odors, but longitudinal measurements in which the tested odors are not repeatedly presented between sessions have instead emphasized remarkable stability of mitral/tufted tuning over days to months across multiple groups. I also think it would strengthen the manuscript to avoid anchoring the empirical comparison too heavily on a single paradigm (Yamada et al., 2017). The experimental literature spans multiple regimes, including daily odor exposure with ongoing experience and longitudinal measurements in which the tested odors are not repeatedly presented between sessions, and these regimes can yield qualitatively different degrees of reorganization. Situating the model explicitly within this broader landscape, rather than emphasizing one dataset, would make the interpretation clearer and prevent readers from overgeneralizing the Yamada findings to baseline bulbar stability. This distinction is especially important because it contrasts with what has been reported in piriform cortex, where representational drift is observed even in the absence of ongoing experience with a given odor set, and where repeated daily encounters with the same odors can slow or arrest that drift.

---

## [Author Response]

The following is the authors’ response to the original reviews.

We thank the editor and reviewers for their thoughtful and constructive feedback. We appreciate that all reviewers recognized the value of our study in linking adult neurogenesis and synaptic plasticity to representational drift in the olfactory system. They described the model as elegant and well-motivated, and agreed that it provides new theoretical insight into how stability and adaptability can coexist in sensory representations. The reviewers also identified areas where our manuscript could be strengthened, and as outlined in our revision plan we have:

(1) Refined our description of mitral/tufted cell stability and expand on within-session and across-day variability.

(2) Substantially expanded the Discussion to compare our modeling assumptions with experimental findings and recent anatomical evidence. Additionally, we have included the limitations of the study and areas for future investigation.

(3) Included a clearer description of the STDP implementation, plastic synapses, and their functional effects.

(4) Add a short section outlining model-based predictions that can guide future experiments. We also made minor textual edits to improve precision and flow, including citing prior conceptual work and clarifying model procedures.

These changes have strengthened both the conceptual framing and technical clarity of the paper. We are grateful for the reviewers’ careful reading and valuable suggestions.

**Public Reviews:**

**Reviewer #1 (Public review):**
Summary:The authors build a network model of the olfactory bulb and the piriform cortex and use it to run simulations and test their hypotheses. Given the model's settings, the authors observe drift across days in the responses to the same odors of both the mitral/tufted cells, as well as of piriform cortex neurons. When representing the M/T and PCx responses within a lower-dimensional space, the apparent drift is more prominent in the PCx, while the M/T responses appear in comparison more stable. The authors further note that introducing spike-time dependent plasticity (STDP) at bulb synapses involving abGCs slows down the drift in the PCx representations, and further link this to the observation that repeated exposure to the same odorant slows down drift in the piriform cortex.The model is clearly explained and relies on several assumptions and observations:(1) Random projections of MTC from the olfactory bulb to the piriform cortex, random intra-piriform connectivity, and random piriform to bulb connectivity.(2) Higher dimensionality of piriform cortex representations compared to M/T responses, which enables superior decoding of odor identity in the piriform cortex.(3) Spike time-dependent plasticity (STDP) at synapses involving the abGCs.The authors address an open topical problem, and the model is elegant in its simplicity. I have however, several major concerns with the hypotheses underlying the model and with its biological plausibility.Concerns:(1) In their model, the authors propose that MTC remain stable at the population level, despite changes in individual MTC responses.The authors cite several experimental studies to support their claims that individual MTC responses to the same odors change (some increase, some decrease) across days. Interpreting the results of these studies must, however, take into account the variability of M/T responses across odor presentation repeats within the same session vs. across sessions. In the Shani-Narkiss et al., Frontiers in Neural Circuits, 2023 study referenced, a large fraction of the variability across days in M/T responses is also observed across repeats to the same odorant in the same session (Shani-Narkiss et al., Figure 4), while the authors have M/T responses in the same session that are highly reproducible. This is an important point to consider and address, since it constrains how much of the variability in M/T responses can be attributed to adult neurogenesis in the olfactory bulb versus to other networks' inhibitory mechanisms, which do not rely on neurogenesis. In the authors' model, the variability in M/T responses observed across days emerges as a result of adult-born neurogenesis, which does not need to be the main source of variability observed in imaging experiments (Shani-Narkiss et al., Figure 4).

We agree with the reviewer and believe this is a critical discussion point. Indeed, both in Shani-Narkiss et al, Kay and Laurent, 1999, and in our lab, we observe trial-to-trial variability that occurs in the same recording session; as the reviewer correctly points out, this cannot be due to neurogenesis. These fluctuations may be trial to-trial noise, or reflect dynamics associated with other behaviors such as running (Chockanathan, et al. 2021) and decision making (Kay and Laurent, 1999). There is growing repertoire of literature showing that neural variability in early sensory coding appears to depend on behavioral fluctuations and internal states (Niell and Stryker for example). This variability that happens within a session in the Shani-Narkiss et al work may reflect some of these behaviorally relevant features of early olfactory coding, something that our model cannot account for. This is an excellent discussion point and we have included text (line 153-157, and line 321-330) in the manuscript to note this aspect of the data and how one can think of it in the context of our results.

Another study (Kato et al., Neuron, 2012, Figure 4) reported that mitral cell responses to odors experienced repeatedly across 7 days tend to sparsen and decrease in amplitude systematically, while mitral cell responses to the same odor on day 1 vs. day 7 when the odor is not presented repeatedly in between seem less affected (although the authors also reported a decrease in the CI for this condition). As such, Kato et al. mostly report decreases in mitral cell odor responses with repeated odor exposure at both the individual and population level, and not so much increases and decreases in the individual mitral cell responses, and stability at the population level.

Thank you for raising this important point regarding the findings of Kato et al. (2012). We agree that their results suggest increased sparsening and stability in M/T cell odor responses with repeated exposure. However, as noted in Yamada et al. (2017), the experimental literature on this question remains mixed. Yamada and colleagues reported a “drastic reorganization of ensemble odor representation” across days and emphasized that “sensory experience does not necessarily cause a major sparsening of the odor response,” explicitly contrasting their findings with those of Kato et al. (2012).

Our model captures the dynamics observed in Yamada et al. (2017), providing a mechanistic explanation for how significant reorganization can emerge in M/T ensembles despite stable low-dimensional population structure. In both Yamada et al (2017) and Kato et al (2012) the investigators have nuanced differences in experimental design (method of head fixation, behavioral paradigm used, training etc.), all of which are known to affect olfactory responses and therefore the degree of sparsity and overlap in population codes. Our model does not include any of these behavioral features that may differentially engage the olfactory circuit and thus affect population responses. Notably, in previous work, we highlight how even simple changes to top down feedback that reflect one phenomenological manipulation to functional connectivity in the olfactory circuit could have disparate effects on the degree of sparsity in neural representations over time whereby this manipulation would be activated by some behavior broadly. In our current model, there is no behavior that would allow us to study the critical features of the neural activity code in the M/T cells. Instead we focus on one specific aspect, adult neurogenesis which we can explicitly manipulate and affect in a biologically meaningful way. The review’s point however is well taken and important, and we have added text to the Discussion (line 336-344) to highlight the differing experimental outcomes and to clarify how our model aligns with the Yamada et al. results.

(2) In Figure 1, a set of GCs is killed off, and new GCs are integrated in the network as abGC. Following the elimination of 10% of GCs in the network, new cells are added and randomly assigned synaptic weights between these abGCs and MTC, GCs, SACs, and top-down projections from PCx. This is done for 11 days, during which time all GCs have gone through adult neurogenesis.Is the authors' assumption here that across the 11 days, all GCs are being replaced? This seems to depart from the known biology of the olfactory bulb granule cells, i.e., GCs survive for a large fraction of the animal's life.

Thank you for raising this important point regarding the lifespan of granule cells (GCs). We agree that developmentally born GCs are not fully replaced. Indeed, multiple studies indicate that some developmentally born GCs can survive for very long periods, up to 18-24 months, essentially the lifetime of the animal (Kaplan, 1985; Petreanu & Alvarez-Buylla, 2002). However, the fraction of total GCs that such long lived GCs constitute remains an open question, in part because of challenges to measure the lifetime survival of newborn neurons. What there is consensus on is the significant size of the granule-cell population undergoing continuous turnover through adult neurogenesis (reviewed in Lepousez et al., 2013).

We should clarify that we do not assume that 100% of the granule cell population turns over in an 11 day period. We use “day” to represent a static epoch over which we can implement plasticity rules across two time scales. Critically, we also randomize the turnover treating every cell in the GC population as equally likely to be replaced. Prior experimental evidence suggests that some GCs are more likely to persist (possibly as a result of experience, Magavi et al., 2005) which may in some regards make our result on stabilization following repeated sensory exposure more dramatic (as the GCs that show the largest change following STDP may also be the ones that are the most stable, and therefore least likely to turnover). We do not include this in our model as we could not identify a framework for “selecting” which GCs would persist that would not be tautological. The point the reviewer raises is critical, and a discussion of these points is warranted - which we now include in the manuscript (line 352-361).

Additionally, there is some evidence that behaviors, such as novelty, can increase the rate of adult neurogenesis (Kamimura et al., 2022, H.van Praag et al.,1999, Gheusi and Lledo., 2014) , suggesting a complex reciprocal relationship between the mechanisms that generate the cells shaping how olfactory stimuli are encoded for and the encoding process itself; our model also does not include any of these dynamic features which represent an additional layer of complexity, which may further provide an intermediate time scale, one of behavioral selection and action, that is slower than the milliseconds on which spike time dependent plasticity happens, but faster than the time scale of neurogenesis. We include this point in the discussion also (line 352-361).

Our 11-day simulation however is designed to uncover how plasticity across multiple timescales (STDP and adult neurogenesis) at the network level shapes odor representations as multiple rounds of GC turnover occur. Changing the timescale and magnitude replacement in the simulations (either in terms of days or percent cells replaced) would affect the degree to which drift happens, but not phenomenon. Additionally, the representational structure in our model at intermediate time points (e.g., days 8~10) would correspond well to scenarios in which some fraction of developmentally born GCs persists in the circuit. Thus, our simulations span a range of possible empirical regimes, from high turnover to partial preservation. We have added discussion to the revised manuscript (line 352-361) clarifying this point and acknowledging the biological heterogeneity in GC lifespans.

(3) The authors' model relies on several key assumptions: random projections of MTC from the olfactory bulb to the piriform cortex, random intra-piriform connectivity, and random piriform to bulb connectivity. These assumptions are not necessarily accurate, as recent work revealed structure in the projections from the olfactory bulb to the piriform cortex and structure within the piriform cortex connectivity itself (Fink et al., bioRxiv, 2025; Chae et al., Cell, 2022; Zeppilli et al., eLife, 2021).How do the results of the model relating adult neurogenesis in the bulb to drift in the piriform cortex representations change when considering an alternative scenario in which the olfactory bulb to piriform and intra-piriform connectivity is not fully distributed and indistinguishable from random, but rather is structured?

Thank you for pointing us to these important studies. We fully agree with the reviewer that the structure of the olfactory system might not be purely random, but we do not believe these papers contradict the level of abstraction used in our model.

Zeppilli et al. (2021) map molecularly defined projection neuron subtypes and their preferential targeting of different cortical and subcortical regions, but they do not report any fine-scale topographic organization of bulb → piriform connectivity that would contradict a view of randomly distributed input to piriform cortex. Studies from our lab using retrograde tracers in the blub show some spatial clustering of piriform cortical neurons whose axons project to the bulb (Padmanabhan et al., 2016, 2019), but these studies do not identify any “functional organization” or structure. Chae et al., (2022) focus on distinct long-range functional loops (mitral ↔ piriform vs tufted ↔ AON) and the differential role of cortical feedback, but again, at the level of cortical regions rather than individual cells and connectivity. Notably, our model does not consider AON.

Finally, Fink et al. (2025) reports a “like-to-like” excitatory connectivity motif within the piriform cortex and an experience-dependent reorganization of inhibitory synapses. As the authors note, “... this like-to-like motif is unlikely to reflect common input from the olfactory bulb”, so it does not conflict with our assumption of broadly random bulb → piriform input. This “like-to-like” motif is reflected in our model by wiring a certain subpopulation of piriform cells. On the other hand, we agree that the experience dependent changes in inhibitory connectivity within PCx are highly relevant for learning related plasticity but fall outside the scope of our study. We intentionally omitted piriform plasticity to isolate the contributions of adult neurogenesis in the bulb and plasticity acting on adult-born granule cells. But incorporating such cortical plasticity is an important direction for future work. We added a discussion (line 395-405) on this important point raised by the reviewer in the revised manuscript.

(4) I didn't understand the logic of the low-dimensional space analysis for M/T cells and piriform cortex neurons (Figures 2 & 3). In the authors' model, the full-ensemble M/T responses are reorganized over time, presumably due to the adult-born neurogenesis. Analyzing a lower-dimensional projection of the ensemble trajectories reveals a lower degree of re-organization. This is the same for the piriform cortex, but relatively, the piriform ensembles displayed in a low-dimensional embedding appear to drift more compared to the M/T ensembles.This analysis triggers a few questions: which representation is relevant for the brain function - the high or the low-dimensional projection? What fraction of response variance is included in the low-dimensional space analysis? How did the authors decide the low-dimensional cut-off? Why does STDP cause more drift in piriform cortex ensembles vs. M/T ensembles? Is this because of the assumed higher dimensionality of the piriform cortex representations compared to the mitral cells?

Thank you for these thoughtful questions. We clarify the logic and purpose of the low-dimensional analyses and address each point below.

(1) Which representation is relevant for brain function, the high-dimensional or low-dimensional one?

We believe both representations are meaningful, with each capturing different aspects of the neural code. The high-dimensional activity reflects the full variability of individual cell responses, while the low-dimensional projection captures the dominant population level components that downstream areas are most likely to use for readout. We found that the low-dimensional representations are more stable in the bulb than in PCx, suggesting that information is used differentially between the two areas. The bulb provides a stable, sensory-anchored population code that reliably represents odor identity over time, consistent with both electrophysiological and behavioral studies (Nagayama et al., 2004, Chen et al., 2009, Davison and Katz, 2007, Cavaretta et al., 2018). This is consistent with its role as the first stage of information processing in the olfactory system which provides faithful representations that downstream circuits receive. The piriform cortex, by contrast, transforms this stable input into a more flexible representation. Drift in its low-dimensional space may reflect ongoing plasticity (Schoonover et al., Nature, 2021), integration of contextual signals, or higherdimensional computations characteristic of PCx (Fink et al., bioRxiv, 2025), suggesting its role more as an associative cortex instead of a pure sensory cortex.

(2) What fraction of variance is included in the low-dimensional space, and how was the cutoff chosen?

In our simulations, these PCs captured the majority of variance relevant for odor identity (~60–70% for M/T cells and ~55–65% for piriform cortex). We now report these fractions explicitly in Methods (line 937-939).

(3) Why does STDP cause more drift in piriform-cortex ensembles than in M/T ensembles? Does this reflect higher dimensionality in piriform cortex?

In our model, STDP does not cause more drift in PCx. It actually reduces drift and stabilizes PCx representations relative to the condition without STDP (as shown in Fig. 4C2). STDP has a much smaller effect in the bulb because: (1) M/T cells continue to receive stable odor input from the glomeruli and (2) the low-dimensional M/T representation is already stable even without plasticity. We have edited the manuscript to reiterate this point in both the results and discussion.

The reviewer is correct that the piriform cortex naturally exhibits more drift than the bulb, and their comment that this is due to its substantially higher representational dimensionality is spot on. The PCx contains many more neurons, receives highly divergent OB → PCx inputs, and has dense recurrent connectivity, all of which create many more degrees of freedom through which representations can drift. Additionally, because individual PCx neurons are sampling from a substantially more diverse combinatorial space of inputs (include feedback to piriform from an array of regions, Illig, 2005, Majak et al., 2004, Chapuis et al., 2013), the “dimensionality” of the population code is likely higher dimensional. While STDP stabilizes the dimensions of the PCx representation that are reinforced during plasticity, due to the large number of orthogonal dimensions available, some residual drift remains. Additionally, as the reviewer notes, there are some forms of plasticity, such as inhibitory plasticity in PCx that are not included in the model, that may also have an impact on both the representations, and the underlying dimensionality of those representations. We include these points in the discussion (line 381-394).

(5) Could the authors comment whether STDP at abGC synapses and its impact on decreasing drift represent a new insight, and also put it into context? Several studies (e.g., Lledo, Murthy, Komiyama groups) reported that abGC integrates in the network in an activity-dependent manner, and not randomly, and as such stabilizes the active neuronal responses, which is consistent with the authors' report.Related, I couldn't find through the manuscript which synapses involving abGCs they focus on, or what is the relative contribution of the various plastic synapses shown in the cartoon from Figure 4 A1 (circles and triangles).

We thank the reviewer for raising this question. As the reviewer pointed out, several studies have shown that abGCs integrate into the bulb circuit in an activity dependent manner. They preferentially form synapses onto mitral/tufted cells that respond to behaviorally important odors, this “selection of surviving cells” is not included in our model. Instead, we use STDP at the synaptic level. This is of course not analogous, but provides a computational framework wherein the selection of surviving abGCs could be incorporated in future studies. It is perhaps notable that in our large scale simulations, synaptic changes at the population level may reflect some of this activity-dependent selection.

To that end, our model provides a new insight and suggests a broader function for adult neurogenesis. For example, when certain odors are reinforced in an activity dependent manner, abGCs born during that period may stabilize the circuits that respond to those odors. The resulting reduction of drift would help keep the representation of those odors stable over time, even while other parts of the circuit continue to change. We now highlight this idea in the Discussion (line 366-373).

For the second part of the question: in our model, STDP acts on two sets of connections. It applies to the synapses onto abGCs from M/T cells, GC/SAC cells, and PCx neurons. It also applies to the synapses that abGCs project to, including those onto M/T cells and GC/SAC cells. We have clarified this in the revised Methods (line 10011004).

(6) The study would be strengthened, in my opinion, by including specific testable predictions that the authors' models make, which can be further food for thought for experimentalists.How does suppression of adult-born neurogenesis in the OB impact the stability of mitral cell odor responses? How about piriform cortex ensembles?

We appreciate the reviewer’s suggestion and formalize the following two predictions from our model:

Prediction 1: Suppressing adult neurogenesis will reduce spontaneous representational drift in the PCx. Increasing spike-timing-dependent plasticity during periods of experience with a specific odor will selectively stabilize representations of that odor.

Prediction 2: Adult neurogenesis will not affect AON representations of odor identity or concentration in the same way that PCx representations are altered and drift.

We include these two ideas in the discussion as experimentally testable predictions.

**Reviewer #2 (Public review):**
Summary:The authors address a critical problem in olfactory coding. It has long been known that adult neurogenesis, specifically in the form of adult-born granule cells that embed into the existing inhibitory networks on the olfactory bulb, can potentially alter the responses of Mitral/Tufted neurons that project activity to the Piriform Cortex and to other areas of the brain. Fundamentally, it would seem that these granule cells could alter the stability of neural codes in the OB over time. The authors develop a spiking network model to explore how stability can be achieved both in the OB over time and in the PC, which receives inputs. The model recapitulates published activity recordings of M/T cells and shows how activity in different M/T cells from the same glomerulus shifts over time in ways that, in spite of the shift, preserve population/glomerular level codes. However, these different M/T cells fan out onto different pyramidal cells of the PC, which gives rise to instability at that level. STDP then, is necessary to maintain stability at the PC level as long as odor environments remain constant. These results may also apply to a similar neurogenesis-based change in the Dentate Gyrus, which generates instability in CA1/3 regions of the hippocampusStrengths:A robust network model that untangles important, seemingly contradictory mechanisms that underlie olfactory coding.Weaknesses:The work is a significant contribution to understanding olfactory coding. But the manuscript would benefit from a brief discussion of why neurogenesis occurs in the first place - e.g., injury, ongoing needs for plasticity, and adapting to turnover of ORNs. There is literature on this topic. It seems counterintuitive to have a process in the MOB (and for that matter in the DG) that potentially disrupts the ability to generate stable codes both in the MOB and PC, and in particular a disruption that requires two different mechanisms - multiple M/T cells per glomerulus in the MOB and STDP in the PC - to counteract.

We appreciate the reviewer’s suggestion and added discussion on this point in the revised manuscript (line 431-435).

Given that neurogenesis has an important function, and a mechanism is in place to compensate for it in the MOB, why would it then be disrupted in fan-out projections to the PC? The answer may lie in the need for fan-out projections so that pyramidal neurons in the PC can combinatorially represent many different inputs from the MOB. So something like STDP would be needed to maintain stability in the face of the need for this coding strategy.This kind of discussion, or something like it, would help readers understand why these mechanisms occur in the first place. It is interesting that PC stability requires that odor environments be stable, and that this stability drives PC representational stability. This result suggests experimental work to test this hypothesis. As such, it is a novel outcome of the research.

We agree with the reviewer. The fan-out from the bulb to the piriform cortex is essential for the combinatorial coding that allows PCx neurons to represent many odor features and mixtures. This architecture gives the piriform cortex great coding capacity, but it also makes the system sensitive to small changes in its inputs. As a result, drift that originates in the bulb can spread more easily in PCx. A stabilizing mechanism is therefore needed downstream. In our model, STDP provides this stabilization by reinforcing the dimensions that carry meaningful odor structure. This allows the piriform cortex to keep a stable population code even when its inputs change over time. Neurogenesis supplies the flexibility, the fan-out supplies the expressive power, and STDP supplies the stability. All three elements work together to support a system that must recognize odors reliably while still adapting to new sensory experiences. We have added discussion on this point in the revised manuscript (line 395-405).

**Reviewer #3 (Public review):**
SummaryThe authors set out to explore the potential relationship between adult neurogenesis of inhibitory granule cells in the olfactory bulb and cumulative changes over days in odorevoked spiking activity (representational drift) in the olfactory stream. They developed a richly detailed spiking neuronal network model based on Izhikevich (2003), allowing them to capture the diversity of spiking behaviors of multiple neuron types within the olfactory system. This model recapitulates the circuit organization of both the main olfactory bulb (MOB) and the piriform cortex (PCx), including connections between the two (both feedforward and corticofugal). Adult neurogenesis was captured by shuffling the weights of the model's granule cells, preserving the distribution of synaptic weights. Shuffling of granule cell connectivity resulted in cumulative changes in stimulus-evoked spiking of the model's M/T cells. Individual M/T cell tuning changed with time, and ensemble correlations dropped sharply over the temporal interval examined (long enough that almost all granule cells in the model had shuffled their weights).Interestingly, these changes in responsiveness did not disrupt low-dimensional stability of olfactory representations: when projected into a low-dimensional subspace, population vector correlations in this subspace remained elevated across the temporal interval examined. Importantly, in the model's downstream piriform layer, this was not the case. There, shuffled GC connectivity in the bulb resulted in a complete shift in piriform odor coding, including for low-dimensional projections. This is in contrast to what the model exhibited in the M/T input layer. Interestingly, these changes in PCx extended to the geometrical structure of the odor representations themselves. Finally, the authors examined the effect of experience on representational drift. Using an STDP rule, they allowed the inputs to and outputs from adult-born granule cells to change during repeated presentations of the same odor. This stabilized stimulus-evoked activity in the model's piriform layer.StrengthsThis paper suggests a link between adult neurogenesis in the olfactory bulb and representational drift in the piriform cortex. Using an elegant spiking network that faithfully recapitulates the basic physiological properties of the olfactory stream, the authors tackle a question of longstanding interest in a creative and interesting manner. As a purely theoretical study of drift, this paper presents important insights: synaptic turnover of recurrent inhibitory input can destabilize stimulus-evoked activity, but only to a degree, as representations in the bulb (the model's recurrent input layer) retain their basic geometrical form. However, this destabilized input results in profound drift in the model's second (piriform) layer, where both the tuning of individual neurons and the layer's overall functional geometry are restructured. This is a useful and important idea in the drift field, and to my knowledge, it is novel. The bulb is not the only setting where inhibitory synapses exhibit turnover (whether through neurogenesis or synaptic dynamics), and so this exploration of the consequences of such plasticity on drift is valuable. The authors also elegantly explore a potential mechanism to stabilize representations through experience, using an STDP rule specific to the inhibitory neurons in the input layer. This has an interesting parallel with other recent theoretical work on drift in the piriform (Morales et al., 2025 PNAS), in which STDP in the piriform layer was also shown to stabilize stimulus representations there. It is fascinating to see that this same rule also stabilizes piriform representations when implemented in the bulb's granule cells.The authors also provide a thoughtful discussion regarding the differential roles of mitral and tufted cells in drift in piriform and AON and the potential roles of neurogenesis in archicortex.In general, this paper puts an important and much-needed spotlight on the role of neurogenesis and inhibitory plasticity in drift. In this light, it is a valuable and exciting contribution to the drift conversation.

We appreciate the reviewer’s comment and thank them for their thoughtful feedback.

WeaknessesI have one major, general concern that I think must be addressed to permit proper interpretation of the results.I worry that the authors' model may confuse thinking on drift in the olfactory system, because of differences in the behavior of their model from known features of the olfactory bulb. In their model, the tuning of individual bulbar neurons drifts over time.This is inconsistent with the experimental literature on the stability of odor-evoked activity in the olfactory bulb.In a foundational paper, Bhalla & Bower (1997) recorded from mitral and tufted cells in the olfactory bulb of freely moving rats and measured the odor tuning of well-isolated single units across a five-day interval. They found that the tuning of a single cell was quite variable within a day, across trials, but that this variability did not increase with time. Indeed, their measure of response similarity was equivalent within and across days. In what now reads as a prescient anticipation of the drift phenomenon, Bhalla and Bower concluded: "it is clear, at least over five days, that the cell is bounded in how it can respond. If this were not the case, we would expect a continual increase in relative response variability over multiple days (the equivalent of response drift). Instead, the degree of variability in the responses of single cells is stable over the length of time we have recorded." Thus, even at the level of single cells, this early paper argues that the bulb is stable.This basic result has since been replicated by several groups. Kato et al. (2012) used chronic two-photon calcium imaging of mitral cells in awake, head-fixed mice and likewise found that, while odor responses could be modulated by recent experience (odor exposure leading to transient adaptation), the underlying tuning of individual cells remained stable. While experience altered mitral cell odor responses, those responses recovered to their original form at the level of the single neuron, maintaining tuning over extended periods (two months). More recently, the Mizrahi lab (Shani-Narkiss et al., 2023) extended chronic imaging to six months, reporting that single-cell odor tuning curves remained highly similar over this period. These studies reinforce Bhalla and Bower's original conclusion: despite trial-to-trial variability, olfactory bulb neurons maintain stable odor tuning across extended timescales, with plasticity emerging primarily in response to experience. (The Yamada et al., 2017 paper, which the authors here cite, is not an appropriate comparison. In Yamada, mice were exposed daily to odor. Therefore, the changes observed in Yamada are a function of odor experience, not of time alone. Yamada does not include data in which the tuning of bulb neurons is measured in the absence of intervening experience.)Therefore, a model that relies on instability in the tuning of bulbar neurons risks giving the incorrect impression that the bulb drifts over time. This difference should be explicitly addressed by the authors to avoid any potential confusion. Perhaps the best course of action would be to fit their model to Mizrahi's data, should this data be available, and see if, when constrained by empirical observation, the model still produces drift in piriform. If so, this would dramatically strengthen the paper. If this is not feasible, then I suggest being very explicit about this difference between the behavior of the model and what has been shown empirically. I appreciate that in the data there is modest drift (e.g., Shani-Narkiss' Figure 8C), but the changes reported there really are modest compared to what is exhibited by the model. A compromise would be to simply apply these metrics to the model and match the model's similarity to the Shani-Narkiss data. Then the authors could ask what effect this has on drift in piriform.The risk here is that people will conclude from this paper that drift in piriform may simply be inherited from instability in the bulb. This view is inconsistent with what has been documented empirically, and so great care is warranted to avoid conveying that impression to the community.

We thank the reviewer for highlighting this important issue. We agree that the interpretation of our model requires care to avoid implying that the olfactory bulb exhibits spontaneous drift. As the reviewer points out, the empirical literature shows that M/T-cell tuning is highly stable for infrequently experienced odors, but can change with daily, persistent odor exposure (e.g., Kato et al., 2012; Yamada et al., 2017).

We thank the reviewer for highlighting the Bhalla and Bower paper, as it is foundational and actually raises a number of interesting and important points. As the authors noted, there was significant variability in trial-to-trial responses over sessions and days in single neurons. This is likely due to on-going dynamics (Laurent, 1999), the impact of behaviorally relevant top-down feedback (Chen and Padmanabhan, 2022), decision making (Kay and Laurent, 1999), and an array of factors that our model does not include. In that manuscript, the authors note “the variability of the same neuron recorded over different days…was not statistically different from the within day comparisons.” While these results appear *prima facie* to be different from our results, there are several reasons why they may not be the case.

First, different metrics are used for measuring neuronal stability, which may contribute to some of the differences. Second, and perhaps more importantly and interestingly, the authors in that study noted the significant trial-to-trial variability within day, which is not present in our study because our model has none of the richness of behavior that Bhalla and Bower found in the freely behaving rat. This variability within day (which is much higher than what we report) would reduce the impact of drift across days - a result that would complicate how plasticity across multiple timescales occurs. We thank the reviewer for the insights on this critical study and include these points in our discussion (line 321-330).

Neural responses to odor representations are incredibly variable across different time scales (Padmanabhan and Urban 2010, Angelo et al 2011, Kapoor and Urban 2006, Friedrich and Laurent, 2001, Smear et al 2011, Wesson et al 2008). In our model, none of this selection of survival related to behavior is included, nor are there specific rules about which synapses may be preferentially strengthened (due to neuro modulation corresponding to behavioral choice and reinforcement learning). Instead, we aimed to recapitulate the experimental design of a few studies (Kato et al 2012, Yamada et al, 2017) to understand how neurogenesis and drift are related. Over the simulated 10 days, the odor is presented every day, and the network is otherwise frozen between sessions—meaning the model lacks mechanisms that would normally support recovery during intervals without odor exposure. Under these conditions, adult neurogenesis effectively interacts with repeated experience, producing gradual changes in individual M/T-cell tuning. Thus, our results should be interpreted as modeling experience dependent changes over the timescale of neurogenesis, not as evidence for spontaneous drift in the bulb. We now state this explicitly in the Discussion to prevent confusion and expand the discussion to incorporate some of these critical ideas (line 321-330).

Major comments (all related to the above point)(1) Lines 146-168: The authors find in their model that "individual M/T cells changed their responses to the same odor across days due to adult-neurogenesis, with some cells decreasing the firing rate responses (Fig.2A1 top) while other cells increased the magnitude of their responses (Fig. 2A2 bottom, Fig. S2)" they also report a significant decrease in the "full ensemble correlation" in their model over time. They claim that these changes in individual cell tuning are "similar to what has been observed by others using calcium imaging of M/T cell activity (Kato et al., 2012 and Yamada et al., 2017)" and that the decrease in full ensemble correlation is "consistent with experimental observations (Yamada et al., 2017)." However, the conditions of the Kato and Yamada experiments that demonstrate response change are not comparable here, as odors were presented daily to the animals in these experiments. Therefore, the changes in odor tuning found in the Kato and Yamada papers (Kato Figure 4D; Yamada Figure 3E) are a function of accumulated experience with odor. This distinction is crucial because experience-induced changes reflect an underlying learning process, whereas changes that simply accumulate over time are more consistent with drift. The conditions of their model are more similar to those employed in other experiments described in Kato et al. 2012 (Figure 6C) as well as Shani-Narkiss et al. (2023), in which bulb tuning is measured not as a function of intervening experience, but rather as a function of time (Kato's "recovery" experiment). What is found in Kato is that even across two months, the tuning of individual mitral cells is stable. What alters tuning is experience with odor, the core finding of both the Kato et al., 2012 paper and also Yamada et al., 2017. It is crucial that this is clarified in the text.

We thank the reviewer. As the issue raised here is related to the previous comment, we have clarified this in the revised text to avoid any misleading comparison and specify what aspects of our computational model map onto experimental studies and what aspects we cannot recapitulate and as a result, the places where our comparisons are limited.

(2) The authors show that in a reduced-space correlation metric, the correlation of lowdimensional trajectories "remained high across all days"..."consistent with a recent experimental study" (Shani-Narkiss et al., 2023). It is true that in the Shani-Narkiss paper, a consistent low-dimensional response is found across days (t-SNE analysis in Shani-Narkiss Figure 7B). However, the key difference between the Shani-Narkiss data and the results reported here is that Shani-Narkiss also observed relative stability in the native space (Shani-Narkiss Figure 8). They conclude that they "find a relatively stable response of single neurons to odors in either awake or anesthetized states and a relatively stable representation of odors by the MC population as a whole (Figures 6-8; Bhalla and Bower, 1997)." This should be better clarified in the text.

We agree with the reviewer that some of the cells in Shani-Narkiss Figure 8B showed relatively stable responses (while others did not). However, there is a clear monotonic increase in the “Average differences” over time, from “Same day” to “1 month” to “6 month”, as quantified in their Figure 8B. Although the author concluded that they "find a relatively stable response of single neurons”, we would argue that their data also provided evidence for what we would term “relatively unstable responses” as found in our model. But per reviewer’s suggestion, we better clarify it in the text now (line 194197).

(3) In the discussion, the authors state that "In the MOB, individual M/T cells exhibited variable odor responses akin to gain control, altering their firing rate magnitudes over time. This is consistent with earlier experimental studies using calcium-imaging." (L3146). Again, I disagree that these data are consistent with what has been published thus far. Changes in gain would have resulted in increased variability across days in the Bhalla data. Moreover, changes in gain would be captured by Kato's change index ("To quantify the changes in mitral cell responses, we calculated the change index (CI) for each responsive mitral cell-odor pair on each trial (trial X) of a given day as (response on trial X - the initial response on day 1)/(response on trial X + the initial response on day 1). Thus, CI ranges from −1 to 1, where a value of −1 represents a complete loss of response, 1 represents the emergence of a new response, and 0 represents no change." Kato et al.). This index will capture changes in gain. However, as shown in Figure 4D (red traces), Figure 6C (Recovery and Odor set B during odor set A experience and vice versa), the change index is either zero or near zero. If the authors wish to claim that their model is consistent with these data, they should also compute Kato's change index for M/T odor-cell pairs in their model and show that it also remains at 0 over time, absent experience.

We appreciate the reviewer’s suggestion and edited the text to make it more accurate (line 319-320).

**Recommendations for the authors:**

**Reviewer #3 (Recommendations for the authors):**
(1) Line 28 "a graduate alteration in sensory perception". We do not know if drift results in changes in perception. If anything, behavioral evidence suggests that perception remains stable in spite of drift. For example, in Driscoll et al. (2017) mice are able to successfully navigate a virtual T maze despite drift, and in Schoonover et al. (2021), mice maintain aversive responses following fear conditioning, despite drift in the piriform. Finally, spatial navigation appears unimpaired despite pronounced drift in the hippocampus (e.g., Climer et al., 2025). It would be more appropriate to say "stimulusevoked activity patterns" than "sensory perception" or other words that refer to neuronal activity rather than cognition or behavior.

We edited the text to make it more accurate per the reviewer’s suggestion (line 27).

(2) In the introduction, the authors state: "This representational drift has led to the hypothesis that PCx, rather than being a primary sensory area, may be more like an association cortical region." (L76-78). However, the hypothesis that PCx operates as an association cortex comes originally from Haberly's work and thinking (e.g., Haberly and Bower, 1984, elaborated in extensive detail in Haberly, 2001). I think it would be appropriate to acknowledge that here.

We added the references to make acknowledge that per the reviewer’s suggestion (line 77).

(3) In the methods, the authors elegantly describe how they induce neurogenesis in their model using weight reshuffling (L805-814). I think it could really help the reader understand the model if this idea were also included in the results section. As the results section currently reads, it seems as if their model implemented neurogenesis in a different fashion: "To do this, following elimination of 10% of the GCs in the network, we added new cells and randomly assigned synaptic weights between these abGCs and M/Ts". I appreciate that in their model, shuffling all the weights of a given GC randomly is akin to "elimination", but I feel like at first blush the results section risks giving an impression a bit different than that actually used in the model.

We edited the text to make it more accurate per the reviewer’s suggestion (line 110-112).